# Covalent organic framework membranes for efficient separation of monovalent cations

Hongjian Wang [1,2], Yeming Zhai[3,4], Yang Li[5], Yu Cao[1,2], Benbing Shi[1,2], Runlai Li [6], Zingting Zhu[1,2], Haifei Jiang[1,2], Zheyuan Guo[1,2], Meidi Wang [1,2], Long Chen [7], Yawei Liu[8], Kai-Ge Zhou[3,4], Fusheng Pan [1,2,4,9,10] & Zhongyi Jiang [1,2,4,9,10]

Covalent organic frameworks (COF), with rigid, highly ordered and tunable structures, can actively manipulate the synergy of entropic selectivity and enthalpic selectivity, holding great potential as next-generation membrane materials for ion separations. Here, we demonstrated the efficient separation of monovalent cations by COF membrane. The channels of COF membrane are decorated with three different kinds of acid groups. A concept of confined cascade separation was proposed to elucidate the separation process. The channels of COF membrane comprised two kinds of domains, acid-domains and acid-free-domains. The acid-domains serve as confined stages, rendering high selectivity, while the acid-free-domains preserve the pristine channel size, rendering high permeation flux. A set of descriptors of stage properties were designed to elucidate their effect on selective ion transport behavior. The resulting COF membrane acquired high ion separation performances, with an actual selectivity of 4.2–4.7 for $K^+/Li^+$ binary mixtures and an ideal selectivity of ~13.7 for $K^+/Li^+$.

Monovalent cations are essential for energy storage materials, of which the global demand is expected to reach a million-ton scale in 2025. Extracting monovalent cations from salt lake, seawater, and brackish water, the major sources of monovalent cations, becomes an imperative demand[1–3]. For instance, $Li^+$ with high purity (>99.5 wt.%) is usually obtained by separating $Li^+$ from ionic mixtures (e.g., 0.003–0.157 wt.% in salt lake and $<2 \times 10^{-7}$ wt.% in seawater) with the coexistence of $K^+$ or $Na^+$. Currently, monovalent cation separations are primarily based on integrated technologies, such as evaporation–precipitation–adsorption, evaporation–extraction, or multi-stage ionic distillation[4]. Membrane technology holds great superiorities in separating monovalent cations owing to the non-thermal, low-carbon footprint attributes. However, the same monovalence, the sub-nanometer ion size (e.g., 1–3 Å for naked ions and 6.5–8 Å for hydrated ions.), as well as the angstrom-sized difference (<2 Å) among different monovalent ions, make it one of the most challenging tasks to separate monovalent cation mixtures[5,6].

Till now, quite few researches involve the separation of mono/monovalent cations. In particular, the overwhelming majority of researches deal with single ion solution and the mono/monovalent

[1]Key Laboratory for Green Chemical Technology of Ministry of Education, School of Chemical Engineering and Technology, Tianjin University, 300072 Tianjin, China. [2]Collaborative Innovation Center of Chemical Science and Engineering (Tianjin), 300072 Tianjin, China. [3]Institute of Molecular Plus, Department of Chemistry, Tianjin University, 300110 Tianjin, China. [4]Haihe Laboratory of Sustainable Chemical Transformations, 300192 Tianjin, China. [5]Department of Chemistry, Tianjin Key Laboratory of Molecular Optoelectronic Science, Tianjin University, 300072 Tianjin, China. [6]Department of Chemistry, National University of Singapore, Singapore 117549, Singapore. [7]State Key Laboratory of Supramolecular Structure and Materials, College of Chemistry, Jilin University, 130012 Changchun, China. [8]Beijing Key Laboratory of Ionic Liquids Clean Process, CAS Key Laboratory of Green Process and Engineering, State Key Laboratory of Multiphase Complex Systems, Institute of Process Engineering, Chinese Academy of Sciences, 100190 Beijing, China. [9]Joint School of National University of Singapore and Tianjin University, International Campus of Tianjin University, Binhai New City, 350207 Fuzhou, China. [10]Zhejiang Institute of Tianjin University, 315201 Ningbo, Zhejiang, China. ✉e-mail: fspan@tju.edu.cn; zhyjiang@tju.edu.cn

cation selectivity is evaluated by the ideal selectivity, which is calculated from the ratio of individual permeation flux of single ion and usually lies within 2–100[7–13]. Few researches deal with binary mono/monovalent cation mixtures using graphene oxide and metal-organic framework membranes[5,14–17], and the acquired selectivity is much lower than the ideal selectivity. For the sake of description simplicity, we define the selectivity for binary ion mixture as actual selectivity. With the emergence of powerful reticular synthesis platform[18–20], a variety of organic framework materials have been developed, which afford unlimited opportunities for discovering innovative and more active mechanisms in monovalent cation separations.

Covalent organic frameworks (COFs) with robust and tunable channels decorated with diverse functional groups are deemed as the disruptive membrane materials for ion separations. The atomically smooth channels of COFs are beneficial for the rapid transport of penetrated ions. The long-range ordered structures impart COF membranes narrow distribution of channel size, ensuring prominent ion sieving effect[21,22]. The channel size of COFs can be finely tailored in the range of 0.5–5.0 nm through the rational screening of linker and linkages[23], which creates a high freedom for membrane construction toward target ion separations. Moreover, COFs possess abundant monodispersed functional groups, thus more easily incorporating multiple physicochemical interactions for high recognition ability toward target ions[24]. These features render COF membranes a wealth of chances for surpassing the trade-off between permeability and selectivity in ion separations. However, because the channel size in COF membranes is usually much larger than the ion sizes, only two reports[25,26] about using COF membranes for mono/bivalent ion separation can be found so far. Therefore, it eagerly awaits great breakthrough to use COF membranes for achieving efficient ion separations.

Herein, we demonstrated the efficient separation of monovalent cations by COF membranes. By designing COFs with different channel size and acid groups and, we proposed a concept of confined cascade separation to elucidate the separation process (Fig. 1): (i) the channels of COF membrane comprised two kinds of domains, the channel domain covered by acid groups, abbreviated as acid domain and the channel domain unoccupied by acid groups, abbreviated as acid-free domain. The acid domain, serving as confined stage, can sufficiently bind water molecules to form hydration shells, narrowing the effective channel size; (ii) all acid domains aligned along the channels of COF membrane can amplify the sieving ability of each confined stage, rendering high selectivity for monovalent cations; (iii) the acid-free-domains preserve the original channel size, conferring high permeability for monovalent cations. Accordingly, we chose a set of

descriptors, including hydration energy, group density (number of acid groups within each pore), group distance, to determine the confined stage properties. The hydration intensity ranged from −0.15 to −0.44 eV, group density ranged from 0 to 6 for each pore, and group distance ranged from 6 Å to 25 Å. Based on the thickness of COF membranes, the stage number varied from ~2000 to ~10,000. Furthermore, our de novo designed COF (TpPa–PO3H2) membrane achieved superior performance in directly sieving the binary monovalent cation mixtures, with an actual selectivity of 4.2–4.7 for K+/Li+ binary mixtures, an ideal selectivity of ~13.7 for K+/Li+.

## Results

### Fabrication and characterization of COF membranes

COF membranes were assembled by nanosheets which were prepared through an oil–water–oil tri-phase interfacial polymerization method (Fig. 2a): (i) a top organic phase containing diamine monomers with three kinds of acid groups, including 2,5-diaminophenylphosphonic acid group (Pa–PO3H2), 2,5-diaminobenzenesulfonic acid group (Pa–SO3H) and 2,5-diaminobenzoic acid group (Pa–CO2H). The monomer design and characterizations were shown in Supplementary Figs. 1–4; (ii) a middle aqueous phase (e.g., pure water, acid or basic solution); iii) a bottom organic phase containing aldehyde monomers (1,3,5-triformylphloroglucinol, Tp). Under static condition, diamine and aldehyde monomers diffused oppositely and encountered in the aqueous phase to trigger reactive assembly of COF nanosheets. The COF models had ideal pristine channels of 14 Å and phosphoric acid (TpPa–PO3H2), sulfonic acid (TpPa–SO3H) and carboxylic acid (TpPa–CO2H) along the 1D channels (Fig. 2b). Using TpPa–PO3H2 as example, the atomic force microscopy (AFM) and high-resolution transmission electron microscopy (HRTEM) images verified that the nanosheets had lamellar structures with the lateral dimension of 2.0–5.0 μm and thickness around 1.1 nm, with the aspect ratio exceeding 1000 (Fig. 2c, d). The crystalline nature of TpPa–PO3H2 nanosheets was confirmed by high-resolution TEM and selected area electron diffraction (SAED) images, where the regularly arranged lattice fringe and diffraction spots can be observed (right insert of Fig. 2d). The scattering vector of diffraction spots was about 0.35 Å−1, matching well with the (100) plane of COFs[25]. Elemental mapping measurement indicated that the P element was evenly distributed within TpPa–PO3H2 nanosheets (Supplementary Fig. 5). TpPa–SO3H and TpPa–CO2H nanosheets also exhibited long-range highly ordered and well-defined morphologies (Supplementary Figs. 6–8 and Supplementary Table 1).

COF nanosheets were assembled into membranes via the filtration method. The stable dispersion and high aspect-thickness ratio of COF

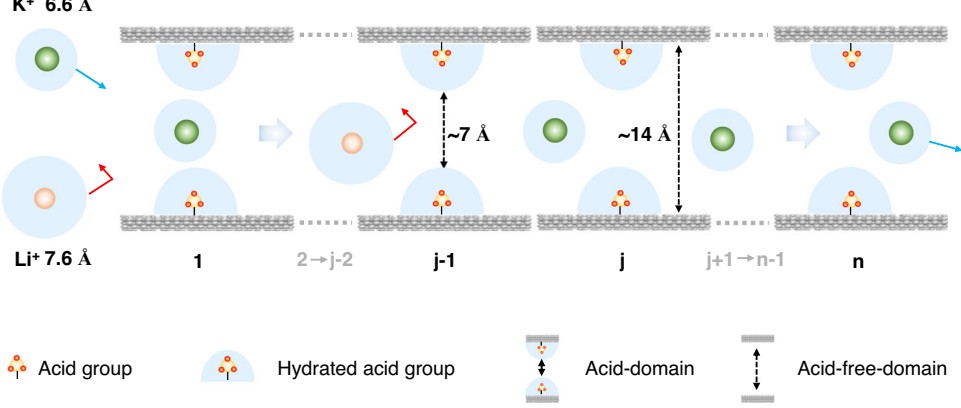

**Fig. 1 | Schematic illustration of confined cascade separation.** The acid domains strongly bind with water molecules to form hydration shells, serving as confined stage to narrow the mass transport channels. The cascade confined stages along the channels contribute to the high-ionic selectivity of membranes (j-2, j−1, j, and j+1 mean the increasing numbers ranging between 1 and n). The acid-free domains confer high permeability of monovalent cations.

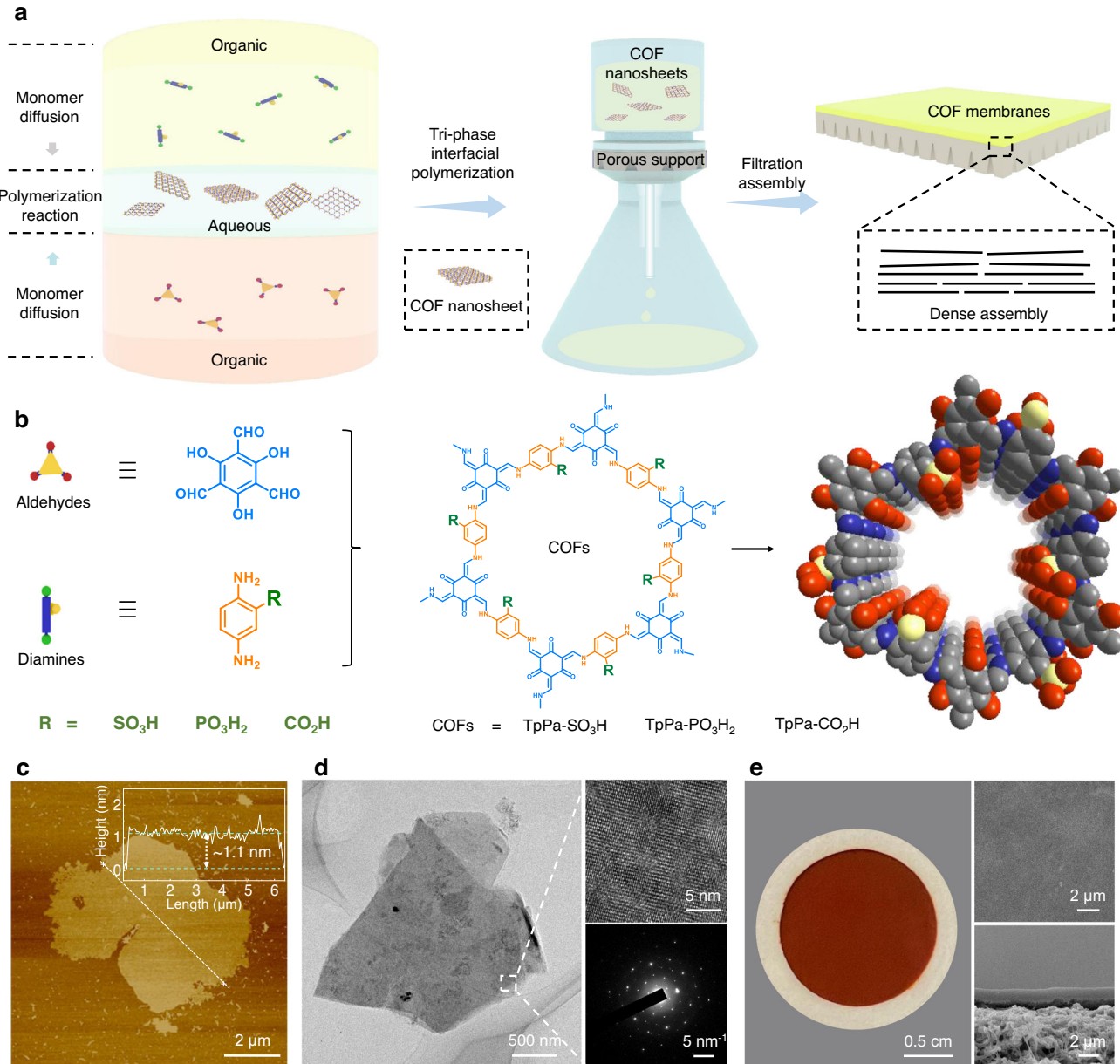

**Fig. 2 | Fabrication and morphology characterization of COF nanosheets and membranes. a** Schematic illustration of the tri-phase interfacial polymerization method to synthesize COF nanosheets and filtration method to fabricate COF membranes. For tri-phase interfacial polymerization, the top organic phase contained diamine monomers with varied acid groups; the middle aqueous phase was comprised of pure water, acid or basic solution; and the bottom organic phase contained aldehyde monomers. After a period of reaction, COF nanosheets were formed in the aqueous phase. COF nanosheets were subsequently filtrated onto porous supports to fabricate COF membranes. **b** Chemical formula of aldehydes and diamines and molecular skeletons of TpPa−PO₃H₂, TpPa−SO₃H, and TpPa−CO₂H used to prepare COF nanosheets and membranes. **c, d** morphologies of the prepared TpPa−PO₃H₂ nanosheets. AFM image in (**c**) demonstrated the high aspect ratio of the TpPa−PO₃H₂ nanosheets. HRTEM images in (**d**) was consistent with AFM results, indicating the well-grown laminar morphology of TpPa−PO₃H₂ nanosheets. The crystalline planes of the nanosheets were also confirmed by SAED measurements (lower-right of **d**). **e** Digital photo and SEM images of TpPa−PO₃H₂ membrane obtained by filtration assembly of TpPa−PO₃H₂ nanosheets, showing uniform and dense morphologies. Source data are provided as a Source Data file.

nanosheets contribute to well-defined structures of COF membranes (Fig. 2e and Supplementary Fig. 9). The top-view of scanning electron microscopy (SEM) images manifested that the TpPa-PO₃H₂ membrane had uniformly dark-red morphologies and no obvious pinholes or cracks. The cross-section view of SEM images manifested that the TpPa-PO₃H₂ membrane had a thickness of 2.0 ± 0.2 μm (tuned by the concentration of casting solution). Interestingly, the dense and uniform morphologies of COF membranes were similar to that of thin (less than 500 nm) graphene oxide membranes[26,27]. Such phenomenon could be attributed to the ultrahigh aspect ratio of COF nanosheets, which rendered plenty of contact sites for adjacent nanosheets.

Moreover, the abundant functional groups on COF nanosheets also intensified the interactions among COF nanosheets.

¹³C solid-state nuclear magnetic resonance (ssNMR, Fig. 3a) showed −C=O and C−N groups at 185 ppm and 125 ppm, which were consistent with Fourier transform infrared spectroscopy (FTIR) result shown in Supplementary Fig. 10, verifying the existence of keto-structure within the skeleton of COF membranes[28,29]. Moreover, the COF membranes showed characteristic signals (−C=O, −C−N, and C−C) on the C 1s region (158−166 eV) of X-ray photoelectron spectroscopy (XPS) measurements, which further revealed the existence of keto-structures (Supplementary Fig. 11), ensuring

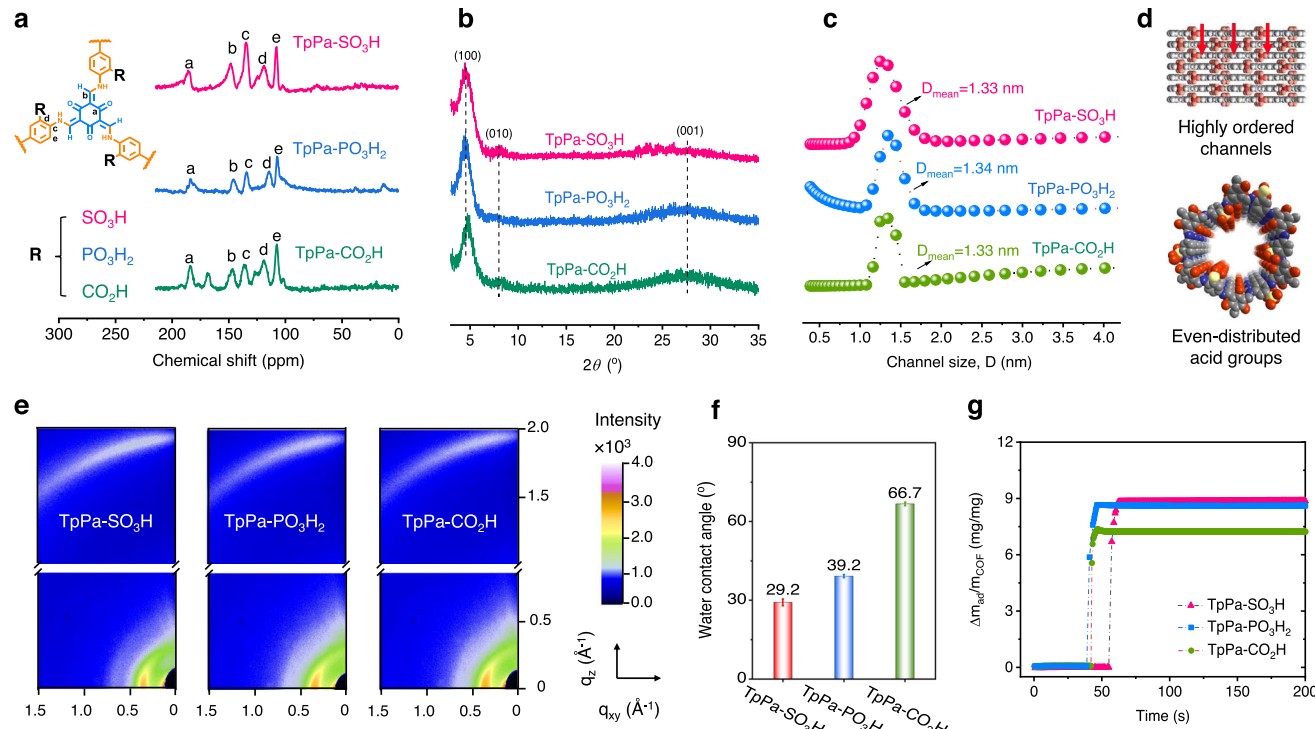

**Fig. 3 | Characterizations of chemical, crystalline, and pore structures of COF membranes. a** $^{13}C$ MAS NMR spectra. The peaks corresponding to C=O and C-N were detected at -185 ppm and 150 ppm for COFs membranes, revealing the existence of keto-structures within the membranes. **b** XRD pattern. Two signals can be observed in 2 theta of 5° and 27–27.5° attributed to crystalline planes of (100) and (001), respectively, manifesting the high crystallinity of COF membranes. **c** Pore size distribution. The average channel size of membranes concentrated on

1.33–1.34 nm. **d** Schematics of COF membranes with TpPa main skeletons. The COF membranes had well-defined 1D channels and even-distributed acid groups along the channels. **e** 2D-GIWAXS spectra. Signal corresponding to (100) plane set near $q_{xy} = 0$ whereas signal referring to (001) plane concentrates on $q_z = 0$, showing the in-plane distribution of 1D hexatomic-ring channel perpendicular to the membrane surface. **f** water contact angle of COF membranes. **g** QCM spectra of COF membranes. Source data are provided as a Source Data file.

imine-transformation and laying the foundation for crystalline structure formation. The density of acid groups was calculated to be 0.8–1.8 at% on the surface of COF membranes. (Supplementary Table 2). The difference of functionalization degree among COF membranes may stem from the uncontrolled formation of partial amorphous skeletons. In addition, Raman spectroscopy of COF membranes demonstrated low-intensity ratio (lying within 0.43–0.78) between model D and model G (Supplementary Fig. 12), which matched well with that of the previous work[30] and indicated highly ordered assembly and low defects within COF membranes. X-ray diffraction (XRD) pattern showed that the COF membranes had three signals appearing at ~5.0°, ~8.0°, and ~27.5° (Fig. 3b), which corresponded to (100) plane, (010) plane and (001) plane, respectively, agreeing well with the simulated patterns of eclipsed projections (Supplementary Fig. 13 and Supplementary Table 3). The high signal intensity of (100) plane around 5.0° manifested that the periodic 1D channel size of COF membranes had a concentration at ~14 Å. Owing to high crystallinity, the experimental channel size obtained from $N_2$ adsorption–desorption experiments was measured to be 13.3–13.4 Å (Fig. 3c and Supplementary Fig. 14), in good accordance with the projected size of ~14 Å, ensuring long-range ordered channels and even-distributed acid groups along the 1D channels (Fig. 3d). Two-dimensional synchrotron radiation grazing incidence wide-angle X-ray scattering (2D-GIWAXS) spectra in Fig. 3e exhibited a diffraction projection signal of (100) plane emerging at $q_{xy} = 0.35$ Å$^{-1}$ and concentrating near $q_z = 0$ Å$^{-1}$, which revealed a certain in-plane distribution of 1D channel perpendicular to the support of COF membranes. Moreover, a weak diffraction spot attributed to (001) plane appeared at around $q_z = 1.9$ Å$^{-1}$ and

concentrated near $q_{xy} = 0$ Å$^{-1}$ unveils the horizontal orientation of π-π stack interlayers and high crystallinity of COF membranes.

The hydrophilicity of COF membranes with different acid groups using water contact angle (WCA), which gave a value of ~29°, ~39°, and ~66° for TpPa–SO₃H, TpPa–PO₃H₂, and TpPa–CO₂H membranes, respectively (Fig. 3f), showing a gradual decrease trend of hydrophilicity. The high hydrophilicity could fortify the interaction between COFs and water molecules. Moreover, the porous crystalline nature afforded more interaction sites between COFs and water molecules, thus contributing to large water adsorption uptake as measured by quartz crystal microbalance (QCM). The water capacity of TpPa–PO₃H₂ and TpPa–SO₃H membranes was over 20% than that of TpPa–CO₂H, reaching 8.4 mg/mg and 8.7 mg/mg, respectively (Fig. 3g). The charged property further supported the water affinity of COF membranes. Zeta potential results in Supplementary Table 4 showed a value of −131.2 ± 1.2 mV, −129.3 ± 3.1 mV, and −64.0 ± 0.3 mV for TpPa–SO₃H, TpPa–PO₃H₂, and TpPa–CO₂H membranes, respectively. The facile dissociation of −SO₃H and −PO₃H₂ groups endowed the membranes with abundant sites to tightly bind water molecules. In contrast, −CO₂H group with weak dissociation showed lower charge property and hydrophilicity. Apart from moderate water affinity and uptake, the COF membranes had excellent aqueous stability, showing a swelling degree less than 5% during the long-term test (Supplementary Table 5).

Simulation on the hydrated acid groups structures was shown in Fig. 4a. Radial distribution function (RDF) was used to describe the atomic density varied with the distance from one particular atom. For the valence charge model, the solid orange line (H atom) exhibited a closer alignment distance than solid blue line (O atom), demonstrating that water molecules formed H-bonds with the acid groups.

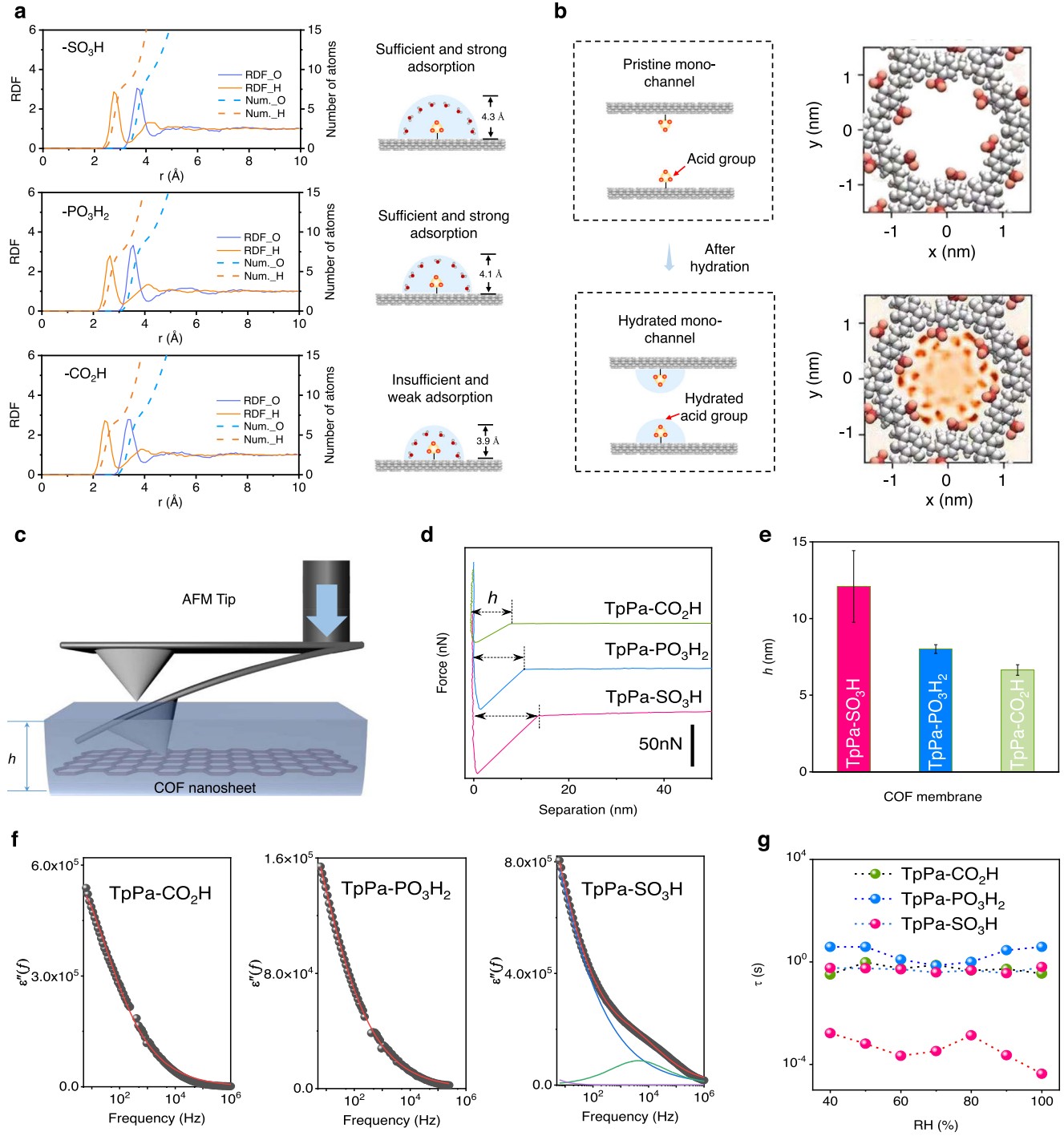

**Fig. 4 | Simulations and characterizations on hydration shells of COF membranes. a** Simulation on the structures of hydrated acid groups. **b** Schematic illustration of hydrated acid groups in COF membranes. **c** Principle of contact AFM measurements for the evaluation of confined water layer and the scheme of jump-to-contact region corresponding to the confined water layer. **d** Force-distance curve of variable COF nanosheets deposited on highly oriented pyrolytic graphite. The separation displacement was defined as piezo height minus cantilever deflection. **e** Length of jump-to-contact region of variable COF nanosheets. The Force-distance curve on 30 different points over $5 \times 5\,\mu m^2$ area were collected for each sample for statistical analysis. **f** Representative dielectric loss spectrum of TpPa–CO$_2$H, TpPa–PO$_3$H$_2$, and TpPa–SO$_3$H collected at 25 °C and 90% RH. The spectrum was fitted by the model of Havriliak–Negami relaxation model, and the fitting results are illustrated by the solid lines. **g** Relaxation time generated from the dielectric loss spectrum of COF membranes. All the error bars in this figure represent the standard deviation of the experiments. Source data are provided as a Source Data file.

Meanwhile, the extreme point of blue line showed the radius of hydration shells (including the size of acid group) for −SO$_3$H, −PO$_3$H$_2$, and −CO$_2$H was about 4.3 Å, 4.1 Å, 3.9 Å, respectively. The peak of the solid blue line, corresponding to the RDF axis, reflected that the interaction intensity between acid groups and water molecules followed the order of −SO$_3$H > −PO$_3$H$_2$ > −CO$_2$H. The dash blue line

attributed to the first derivative of RDF for O atoms was used to describe the number of adsorbed water molecules, which followed the trend of −SO$_3$H ≥ −PO$_3$H$_2$ > −CO$_2$H. For individual channel, the acid group (−SO$_3$H and −PO$_3$H$_2$) could strongly bind water molecules, forming hydration shells to narrow the effective channel size (Fig. 4b). To understand the hydration shells in COF membranes, we employed

two methods to characterize the confined water layers, including AFM and dielectric spectra. Due to the capillary attraction with the confined water on surface[31,32], AFM tip had a sharp acceleration during the contact with the sample (Fig. 4c). Consequently, a jump-to-contact region (h) corresponding to confined water layers could be observed in the force-distance curve (Fig. 4d). Compared with TpPa–CO₂H nanosheets, the jump-to-contact region of TpPa–SO₃H and TpPa–PO₃H₂ membranes were 78.8% and 18.5% longer, respectively (Fig. 4e). Therefore, the thickness of water layer confined on different COF nanosheet followed the order of TpPa–SO₃H > TpPa–PO₃H₂ > TpPa–CO₂H, which was consistent with the simulation results. To gain further insight into the effect of acid groups on the confined water layer, we used dielectric spectra to explore the degree of water molecules confined in COF membranes. Different from dielectric response of bulk water in GHz to THz range, water molecules in confined space showed a response in lower frequency range owing to the constrain interaction[33–35]. Taking the dielectric loss spectra of COF membranes collected at 25 °C and 90% relative humidity (RH) as an example, the TpPa–PO₃H₂ and TpPa–CO₂H membranes showed a single dielectric response below 10 Hz, however, the TpPa–SO₃H membrane exhibited an addition dielectric response in higher frequency region (Fig. 4f). By fitting the dielectric spectra with the Havriliak–Negami relaxation model, the relaxation time of COF membranes under different humidity were quantitatively summarized in Fig. 4g. All the membranes exhibited a slower relaxation time, $\tau_1$ around 0.1–10 s, while a second Havriliak–Negami term was observed in TpPa–SO₃H membrane with a much quicker dipolar relaxation, $\tau_2$ near $10^{-4}$ s. The apparency of $\tau_2$ indicated the formation of second water layer, which was also discovered in clay laminate[36]. The second water layer further verified the increase of water thickness observed in simulations and contact AFM measurements.

## Selective transport of monovalent cations

The monovalent cation transport behavior of COF membranes was evaluated using concentration-driven configuration (Supplementary Fig. 15). 0.1 M aqueous salt solutions, including RbCl, KCl, NaCl, and LiCl were placed in the feed side and deionized water was employed in the permeate side. COF membranes exhibited stable permeation/diffusion of monovalent cations during operation (Supplementary Fig. 16). For simplicity, we abbreviated the hydrated cations as cations in the subsequent description. The ionic diffusion coefficients of TpPa–PO₃H₂ and TpPa–SO₃H membranes were found to strictly depend on the ionic diameter, which had a sharp size cutoff of 6.6 Å (Fig. 5a). When ionic diameter was less than 6.6 Å (in case of Rb⁺ and K⁺), the diffusion coefficients of TpPa–PO₃H₂ and TpPa–SO₃H membranes exceeded $1.2 \times 10^{-8}$ cm² s⁻¹. When ionic diameter was higher than 6.6 Å, the diffusion coefficients of Na⁺ and Li⁺ for TpPa–PO₃H₂ and TpPa–SO₃H membranes were below $1.6 \times 10^{-9}$ cm² s⁻¹, which were about eight times lower than those of Rb⁺ and K⁺. In sharp contrast, the diffusion coefficients of TpPa–CO₂H membrane did not significantly change with the varied ionic diameter, showing $(5.7 \pm 0.81) \times 10^{-9}$ cm² s⁻¹ for Rb⁺, $(7.6 \pm 1.1) \times 10^{-9}$ cm² s⁻¹ for K⁺, $(6.3 \pm 1.8) \times 10^{-9}$ cm² s⁻¹ for Na⁺ and $(4.2 \pm 1.1) \times 10^{-9}$ cm² s⁻¹ for Li⁺. Moreover, the ionic diffusion coefficients for three COFs membranes were four orders of magnitude lower than those of bulk solution. This was mainly due to the large mass transport resistance of monovalent cations across membranes.

We attributed the selective transport of different monovalent cations to the cascading hydrated acid groups along 1D channels of COF membranes, as shown in Fig. 5b. After immersing into aqueous solution, the acid domains with –SO₃H and –PO₃H₂ groups, serving as confined stage, can sufficiently bind water molecules to form hydration shells, and narrowed the effective channel size lying between target cations. Subsequently, thousands of the acid domains along

channels of COF membrane, amplified the sieving ability of each stage, rendering high selectivity for monovalent cations. Meanwhile, the acid-free domains preserved pristine channel size, which was larger than all cations, thus conferring high permeability for monovalent cations. We designed a set of descriptors, i.e., hydration energy, group density, group distance, to determine the stage properties. The hydration energy ranged −0.15 to −0.44 eV, group density ranged 0–6 for each channel, and group distance ranged 6–25 Å. Based on the thickness of COF membranes, the stage number was varied from -2000 to -10,000.

The effect of hydration energy on selective transport of monovalent cations in a confined stage was investigated. By the density functional theory (DFT) method, the hydration energy of –SO₃H, –PO₃H₂, and –CO₂H groups in COF membranes was calculated to be −0.44 eV, −0.36 eV, and −0.15 eV, respectively. With increasing the hydration energy, the permeation rate of K⁺ slightly changed, ranging from 0.14 mol h⁻¹ m⁻² to 0.33 mol h⁻¹ m⁻² (Fig. 5c and Supplementary Fig. 17). In contrast, the permeation rate of Li⁺ sharply decreased from 0.077 mol h⁻¹ m⁻² to 0.013 mol h⁻¹ m⁻², and the corresponding ideal selectivity of K⁺/Li⁺ sharply increased from 1.82 to 16.2. The transport behavior difference between K⁺ and Li⁺ was attributed to the difference of hydration shell in acid domains. Theoretically, the higher the hydration energy was, the more easily the groups bind water molecules to form hydration shells. Therefore, for the –SO₃H and –PO₃H₂ groups with higher hydration energy, abundant (adsorbing 7–9 water molecules for each group) hydration shell and larger hydration shell thickness (4.1–4.3 Å) could be formed in acid domains, which further reduced the channel size lying between the size of K⁺ and Li⁺, affording the selective transport of monovalent cations. However, -CO₂H with lower hydration energy could not bind sufficient water molecules and exhibited a weaker effect on the confined selective transport of monovalent cations.

The acid group density in each confined stage was tuned to assess its effect on selective ion transport. Employing the sulfonic acid COFs as example, the channel size was kept identical while the group density in each channel was varied. When the number of theoretical acidic groups (here called group density) in each channel increased from 0 to 6, the permeation rates of K⁺ and Li⁺ showed an overall decreasing trend, while the corresponding ideal selectivity gradually increased (Fig. 5d and Supplementary Fig. 18). In the absence of acid groups, the permeation rates of K⁺ and Li⁺ lied in a high-value range, which was $0.32 \pm 0.01$ mol h⁻¹ m⁻² and $0.17 \pm 0.03$ mol h⁻¹ m⁻², respectively. In this case, the ideal selectivity of COF membranes for monovalent ions was very low. This indicated that no hydration shell could be formed to leverage selective transport of monovalent cations. When the group density reached 6.0, the selectivity of the membrane was about 18.6, slightly higher than the selectivity (-16.4) of the membrane with group density of 3.0. This may be due to that the increased number of acid groups fortified the confinement effect of each stage, and thus increased the overall ideal selectivity.

The effect of group distance on ion transport performances was subsequently evaluated. By tuning the length and geometry of diamine linkers, we controlled the group distance of –SO₃H at 6.0 Å, 14 Å, and 25 Å (Supplementary Fig. 19). When the group distance was 6.0 Å, the permeation rates of K⁺ and Li⁺ were $0.0089 \pm 0.005$ mol h⁻¹ m⁻² and $0.0093 \pm 0.001$ mol h⁻¹ m⁻², respectively (Fig. 5e). The transport of cations was hindered by COF channels and no obvious ideal selectivity could be acquired. This phenomenon was ascribed to that the smaller channel could not confer enough space for monovalent cations to transport. When the group distance was 14 Å, the calculated ideal selectivity was about 16.4. In this case, the effective channel size narrowed by the hydrated acid groups appropriately permitted the transport of K⁺ and simultaneously blocked the entrance of Li⁺. When the group distance was 25 Å, the permeation rates of both K⁺ and Li⁺ increased

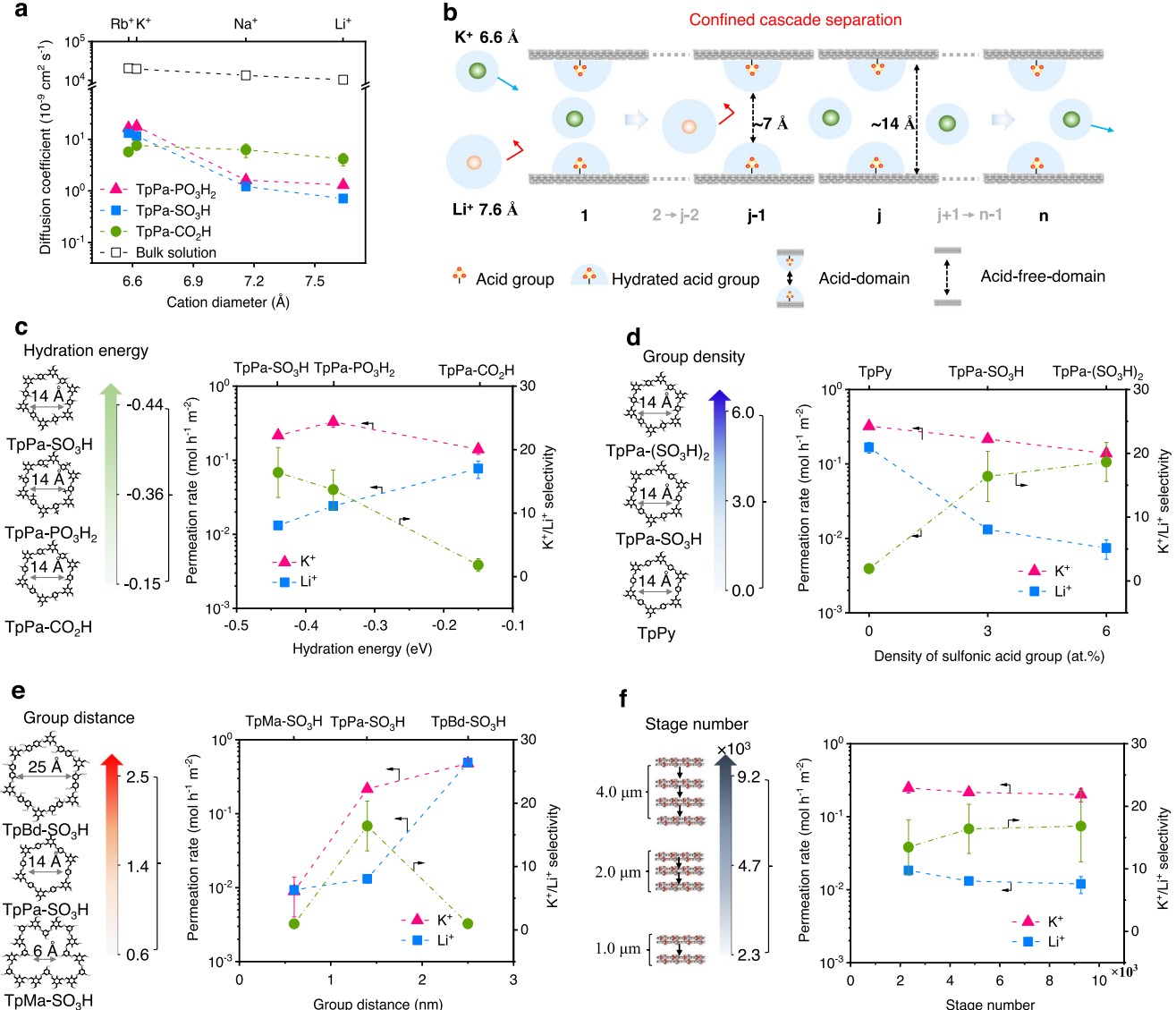

**Fig. 5 | Selective transport of monovalent cations through COF membranes.** **a** Diffusion coefficient of the monovalent cation with a different diameter across TpPa–PO₃H₂, TpPa–SO₃H, and TpPa–CO₂H membranes. Chloride with a concentration of 0.1 M was used as a counterion in all cases. The diffusion coefficients for TpPa–PO₃H₂ and TpPa–SO₃H membranes had a sharp cutoff between the diameter of 6.6 and 7.2 Å. **b** Schematic illustration of the proposed confined cascade separation within COF membranes for selective ion transport. The cascade acid domains conferred selectivity for monovalent cations and the acid-free-domains rendered high permeability of monovalent cations. **c** Transport performances of COF membranes under different hydration energy of confined stage. Overall, increasing the hydration energy could increase the ideal K⁺/Li⁺ selectivity of COF membranes. **d** Transport performances of COF membranes under different group density of confined stage. Increasing the group density brought about the increase of ideal K⁺/Li⁺ selectivity and decrease of permeation rates. **e** Transport performances of COF membranes under different group distance of confined stage. The smaller group distance (-6 Å) and the larger group distance (-25 Å) had no obvious sieving performances for ion transport. In contrast, -14 Å of group distance was found appropriate for selective ion transport. **f** Transport performances of COF membranes under different stage number. Increasing stage number led to the elevated ideal K⁺/Li⁺ selectivity. All the error bars in this figure represent the standard deviation of the experiments. Source data are provided as a Source Data file.

rapidly and exceeded 0.48-0.49 mol h⁻¹ m⁻². The corresponding ideal selectivity only ranged within 1.0–1.1. The experimental result further demonstrated the dominated effect of hydration channel size on selective transport of cations, i.e., too large channel (-25 Å) had no rejection to both of cations, while too small channel (-6 Å) rejected both of cations, and only moderate channel size (-14 Å) could render selective transport of monovalent cations.

The influence of the stage number on ion transport behavior was also investigated. Taking the TpPa–SO₃H membrane as an example, the membrane thickness was tuned at 0.98 μm, 2.0 μm, and 3.9 μm, respectively, to obtain stage number of 2333, 4761, and 9267, respectively (the single-stage thickness was 0.42 nm). As shown in Fig. 5f, with stage number increasing from 2333 to 9267,

the permeation rates of all cations showed a decreasing trend, while the ideal selectivity of K⁺/Li⁺ increased from -13.5 to -16.8. This indicated that the higher the stage number was, the easier it was to amplify the sieving ability of confined stage along 1D COFs channels, and thus achieved selective transport of monovalent cations. Furthermore, the K⁺ permeation rate of COFs membrane was over 0.20 mol h⁻¹ m⁻² as stage number ranging from 2333 to 9267. The acid-free domains retained pristine channel size of -14 Å, thus endowing monovalent cations with high permeability. Furthermore, other factors, such as ionic strength and electrostatic interaction between acid groups and cations, were also investigated to demonstrate selective transport of monovalent cations across COF membranes (Supplementary Figs. 20 and 21 and Supplementary

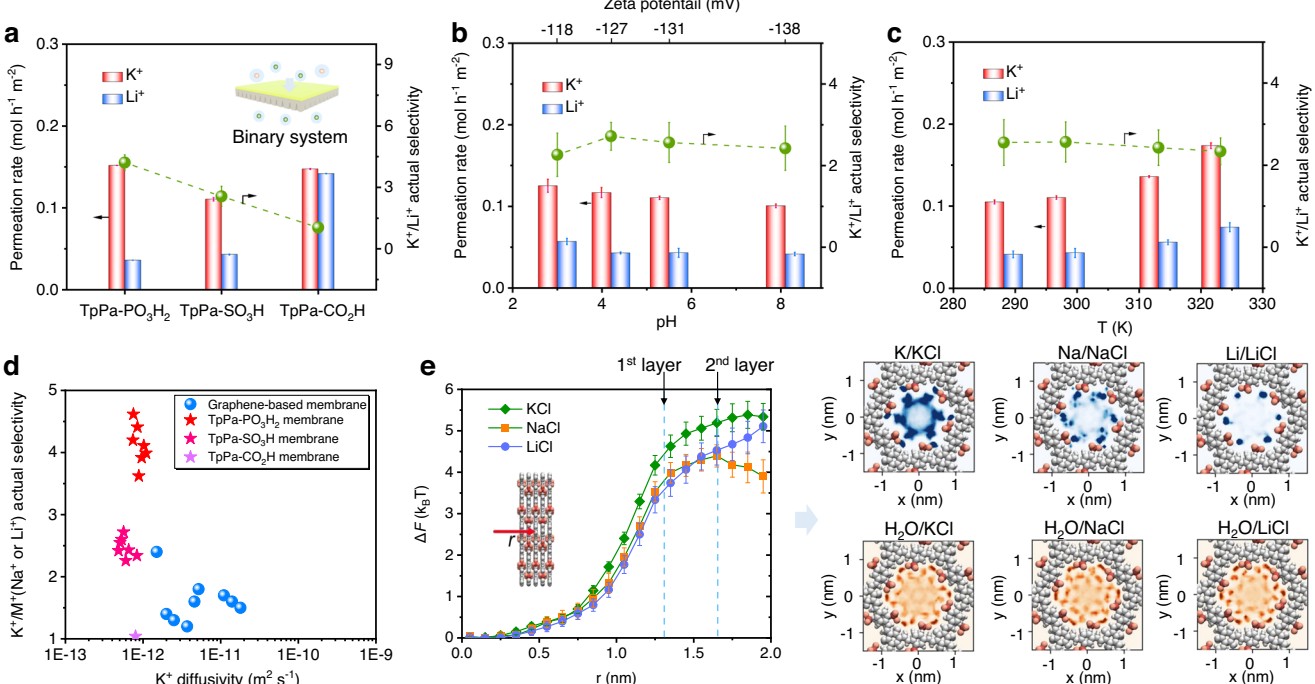

**Fig. 6 | Separation performances for binary mixtures of monovalent cations through COF membranes. a** Separation performances of TpPa–PO$_3$H$_2$, TpPa–SO$_3$H, and TpPa–CO$_2$H membranes for binary mixture of monovalent cations. **b** Separation performances of TpPa–SO$_3$H membranes toward binary mixtures of monovalent cations under different pH. **c** Separation performances of TpPa–SO$_3$H membranes toward binary mixtures of monovalent cations under different operation temperatures. **d** Comparison among COF membranes in this study and other representative membranes in literature toward separating binary mixtures of monovalent cations under concentration-driven configuration. **e** Calculated potential mean force (PMF) along the channels, and probability distribution of various anions and water molecules in TpPa–SO$_3$H channels. All the error bars in this figure represent the standard deviation of the experiments. Source data are provided as a Source Data file.

Table 6). We found that ionic strength and electrostatic interactions did not have a profound effect on the permeation rates for monovalent cations.

## Separation performance for binary mixtures of monovalent cations

Moreover, we used the TpPa–PO$_3$H$_2$, TpPa–SO$_3$H, and TpPa–CO$_2$H membranes to separate binary mixtures of monovalent cations. Under concentration-driven configuration, we set the feed side with a mixtures of 0.1 m KCl and 0.1 m LiCl aqueous solution and the permeation side with deionized water. Figure 6a showed the separation performance of TpPa–PO$_3$H$_2$, TpPa–SO$_3$H, and TpPa–CO$_2$H membranes for binary mixtures of monovalent cation. Compared with higher permeation rate of single cation transport, the permeation rate of binary mixtures had a sharp decrease, which was less than 0.16 mol h$^{-1}$ m$^{-2}$. This was due to the strong coupling effect (hard-core interactions and electrostatic Coulombic potential) between mixed cations and their competition for occupying effective mass transport channels, leading to the increase of permeation energy barrier of each monovalent cation. The actual selectivity of TpPa–CO$_2$H membrane was 1.04 ± 0.16, which was close to its ideal selectivity, demonstrating that TpPa–CO$_2$H membrane had no obvious separation performance toward binary mixtures. In sharp contrast, the actual selectivity of TpPa–PO$_3$H$_2$ and TpPa–SO$_3$H membranes was 4.21 ± 0.37 and 2.56 ± 0.49, respectively. Although the TpPa–PO$_3$H$_2$ and TpPa–SO$_3$H membranes could achieve the efficient separation of binary mixtures of monovalent cations, their actual selectivity was less than their ideal selectivity (about 13.7 and 16.4, respectively). We attributed such phenomenon to the strong coupling effect between binary cations, which limited the hydration of acid domains and led to that the actual selectivity was lower than the ideal selectivity.

We further evaluated the influence of pH on membrane performances. When the pH value was 3.0, 4.2, 5.5 and 8.1, the corresponding zeta potential of TpPa·SO$_3$H membrane was measured to be −118 mV, −127 mV, −131 mV, and −138 mV, respectively. As shown in Fig. 6b and Supplementary Fig. 22a, with the increase of pH from 3.0 to 8.1, the permeation rates of K$^+$ and Li$^+$ in binary system slightly decreased from 0.125 mol h$^{-1}$ m$^{-2}$ and 0.057 mol h$^{-1}$ m$^{-2}$ to 0.101 mol h$^{-1}$ m$^{-2}$, and 0.041 mol h$^{-1}$ m$^{-2}$, respectively. The actual selectivity kept steady and ranged within 2.26–2.71. This result indicated that enhancement in the dissociation of acid groups had slight change on ion separation performances, revealing that the electrostatic interactions was not the main factor during separation. We investigated the effect of operation temperature on separation performances (Fig. 6c and Supplementary Fig. 22b). It could be observed that the actual selectivity had only a decrease with the temperature increasing from 293 to 323 K. Because the increased temperature impaired hydration ability of acid groups, the actual selectivity of membranes was therefore restricted. Compared with the TpPa–PO$_3$H$_2$ membrane, the TpPa–SO$_3$H membrane showed a more obvious change on permeation rates with the varied temperature. This was primarily due to the activity of −SO$_3$H group was more sensitive than −PO$_3$H$_2$ group upon temperature change (Supplementary Fig. 22c, d). Furthermore, we compared the separation performances of COF membranes and other membranes[15,16] for binary mixtures of monovalent cations (Fig. 6d). The TpPa–PO$_3$H$_2$ membrane had a higher performance in actual selectivity over 4.0 for binary monovalent cations.

To elucidate the ion transport process in COF membrane, the potential mean force (PMF) of anion (chloride ion) for KCl, NaCl, and LiCl passing through TpPa–SO$_3$H channels was calculated. As shown in Fig. 6e, the chloride ions need to overcome a certain free energy barrier to enter COF channels due to the electrostatic repulsion

between channels and chloride ions. However, the free energy barrier of chloride ions for KCl, NaCl and LiCl ranged at about 4.0–5.5 times of $k_B T$ and had an inappreciable difference, indicating that the selective transport of cations is not dominated by the electrostatic repulsion effect of their counterions. To gain insight into the separation process, we simulated the probability distribution of monovalent cations and water molecules through COF channel. It was found that the distribution density of K$^+$, Na$^+$, and Li$^+$ in channel gradually decreased (up-right of Fig. 6e), illustrating that K$^+$ was easier to enter and pass through the TpPa–SO$_3$H channel to achieve high permeation rate. This phenomenon was consistent with the trend of K$^+$ permeation in our experimental results. The bottom-right of Fig. 6e showed the distribution behavior of water molecules in the TpPa–SO$_3$H channels. It can be observed that water molecules formed an annular conformation around phosphate ion groups under the interaction of hydrogen bonds. The robust first hydration layer effectively reduced the channel size to about 7.0 Å, which lay within K$^+$ and Na$^+$ (or Li$^+$).

## Discussion

In summary, we demonstrated the efficient separation of monovalent cations by COF membranes. Based on the COF membranes with different channel size and acid groups, we proposed a concept of confined cascade separation to elucidate and intensify the ion transport and separation process. Specifically, we chose a set of descriptors, including hydration energy, group density (number of acid groups within each pore), group distance, to determine and optimize the confined stage properties. By manipulating the channel properties of COF membranes, our de novo designed phosphoric COF (TpPa–PO$_3$H$_2$) membrane exhibited high separation performance for monovalent cations, with an actual selectivity of 4.2–4.7 for K$^+$/Li$^+$ binary mixtures and an ideal selectivity of ~13.7 for K$^+$/Li$^+$. Our work is the pioneering exploitation in separating monovalent ionic mixtures by using organic framework membranes through confined cascade separation.

## Methods

### Materials and chemicals

1,3,5-Triformylphloroglucinol (Tp, 98%) was supplied by Jilin Chinese Academy of Sciences Yanshen Technology Co., Ltd. 2,5-Diaminobenzoic acid (Pa-CO$_2$H, 97%) and 2-bromobenzene−1,4-diamine (97%) were bought from Bidepharmatech Co., Ltd (Shanghai). 2,5-Diaminobenzenesulfonic acid (Pa-SO$_3$H, 98%), trimethylamine (99%), acetonitrile (99.8%), dichloromethane (99%) and acetic acid (99.5%) were purchased from Meryer Co., Ltd. (Shanghai). 4,4′-Diamino-2,2′-biphenyldisulfonic Acid (Bd-SO$_3$H, 70%) and 2,4-diaminobenzenesulfonic acid (Ma-SO$_3$H, 98%) and 2,5-diaminopyridine (Py, 97%) were supplied from Aladdin Co., Ltd. (Shanghai). 2,5-diaminobenzene-1,4-disulfonic acid (Pa-(SO$_3$H)$_2$, 97%), diethyl phosphite (98%), tetrakis(triphenylphosphine)palladium (95%), trimethylsilyl bromide (95%) were obtained from Heowns Biochem Technology Co., Ltd. (Tianjin). Methylbenzene (AR), dimethylformamide (DMF, AR) was bought from Jiangtian Chemicals (Tianjin). Methanol (99.8%) was purchased from Shanghai Macklin Biochemical Co., Ltd. Potassium carbonate (K$_2$CO$_3$, AR), potassium chloride (KCl, GR), sodium chloride (NaCl, AR), lithium chloride (LiCl, AT), and rubidium chloride (RbCl, AR) were obtained from Sigma-Aldrich (Shanghai). Polytetrafluoroethylene (PTFE) support with an average pore size of 0.22 µm was purchased from Haiyan Xindongfang Suhua Technology Co., Ltd. All chemicals were used without further purification.

### Synthesis of COF nanosheets

COF nanosheets were prepared through a tri-phase interfacial polymerization method, which used 50 mL of DMF containing diamine monomers as the top phase, 20 mL of the aqueous solution as the buffer phase, and 80 mL of dichloromethane containing aldehyde monomers as the bottom phase. All the synthesis conditions used 0.1 mmol of Tp as the aldehyde monomers to be dissolved in the bottom phase. The recipes of top phase and buffer phase for nanosheets were illustrated as follows: for sulfonic acid COF nanosheets, 0.15 mmol of diamine (Pa-SO$_3$H, Pa-(SO$_3$H)$_2$, Bd-SO$_3$H, and Ma−SO$_3$H) was added into the bottom phase and 3 M acetic acid was selected as the buffer phase; for phosphoric acid COF nanosheets, 0.15 mmol of diamine (Pa−PO$_3$H$_2$) was added into the top phase and 0.5 M sodium bicarbonate aqueous solution was selected as the buffer phase; for carboxylic acid COF nanosheets, 0.15 mmol of diamine (Pa−CO$_2$H) was put into the top phase while and 0.05 M NaOH aqueous solution was selected as the buffer phase; for TpPy nanosheets, 0.15 mmol of 2,5-diaminopyridine (Py) was dissolved in the top phase and aqueous solution containing 0.5 M acetic acid was screened as the buffer phase. Subsequently, the reaction was carried out with setting the prepared phases in the order of bottom to up and kept under static condition at 25 °C for 3 days. The resultant system was dropped into 20 mL of deionized water and subsequently partitioned to obtain the aqueous colloidal. With dialysis and purification in deionized water for 3 days, the transparent aqueous colloidal of COF nanosheets was successfully obtained.

### Preparation of COF membranes

COF membranes were prepared using filtration assembly method. Briefly, a certain of COF nanosheet colloidal (0.1–0.3 mg mL$^{-1}$) was diluted to 30 mL using deionized water. The solution was then penetrated over porous supports (PTFE with an average pore size of 200 nm) to form the COF membranes. The prepared sulfonic acid, phosphoric acid and carboxylic acid were respectively immersed into 0.1 M sulfonic acid, phosphoric acid and carboxylic acid aqueous solution for 24 h. In this study, we deposited COF nanosheets on PTFE supports for the tests of ion transport. The filtration assembly apparatus was 2.0 cm of filtered diameter and the pressure of filtration was controlled under 0.75 bar. The obtained membranes were cut into a size of 1.4 cm in diameter and at least four membranes with the same recipe were tested to ensure the reproducibility and performance.

### Ion permeation tests of COF membranes

Ion permeation measurements were carried out using a diffusion counter (diffusion cell) apparatus. For testing the selective transport of monovalent cations, 200 mL of 0.1 M chlorate salts, including RbCl, KCl, NaCl, and LiCl were individually added into the feed side, and 200 mL of deionized water was added into the permeate side. For separating the binary mixtures of monovalent cations, 200 mL of chlorate salt mixtures containing 0.1 M KCl and 0.1 M LiCl were set into the feed side and 200 mL of deionized water was added into the permeate side. Both the feed and permeate sides were stirred magnetically to minimize the influence of concentration polarization. After permeation for a period of time under room temperature, the conductivity of permeate side was measured using a conductivity meter (DDS-11A) and the cationic concentration in permeate side was detected by ion chromatography (ICS-600). The permeation rate ($J_i$, mmol h$^{-1}$ m$^{-2}$) of cations was calculated using Eq. (1):

$$J_i = (C \times V)/(M_r \times A \times \triangle t) \qquad (1)$$

where $C$ (µg/mL) was the cation concentration in permeated side, $V$ (mL) was the effective volume of the solution in permeated side ($V$ was 200 mL in this work), $M_r$ (g mol$^{-1}$) was the relative molecular mass of cation, $A$ (cm$^2$) was the effective permeated area, and $\triangle t$ (h) was the permeated time.

In addition, the diffusion coefficients of monovalent ions ($D$, cm$^2$ s$^{-1}$) in COF membranes were calculated based on the classical

law of diffusion as shown in Eq. (2):

$$J_i = \frac{D}{d} \times \triangle c \times \frac{A_{eff}}{A} \qquad (2)$$

where $J_i$ (mol h$^{-1}$ m$^{-2}$) was the permeation rate of monovalent ions, $d$ (μm) was the thickness of COF membranes, $\triangle c$ (mol L$^{-1}$) was the concentration gradient across the membranes, $A$ (cm$^2$) was the membrane area and $A_{eff}$ (cm$^2$) was the pore area for each COF membrane.

The ideal or actual selectivity of COF membranes using Eq. (3):

$$\alpha_{ij} = \frac{j_i/\triangle c_i}{j_j/\triangle c_j} \qquad (3)$$

where $J_i$ (mol h$^{-1}$ m$^{-2}$) and $J_j$ (mol h$^{-1}$ m$^{-2}$) were the permeation rate of cation i and j, respectively; $\triangle c_i$ (mol L$^{-1}$) and $\triangle c_j$ (mol L$^{-1}$) were the concentration gradient of cation i and j, respectively.

Besides, based on the permeation rate ($J_i$, mmol h$^{-1}$ m$^{-2}$) of cations and osmotic pressure ($\triangle p$, bar), the permeability ($P_i$, mmol h$^{-1}$ m$^{-2}$ bar$^{-1}$) of cations could be obtained using Eq. (4):

$$P_i = \frac{J_i}{\triangle p} \qquad (4)$$

## Molecular simulations

We studied the transport of different ions through several layers of COFs using all-atomistic molecular dynamics (MD) simulations. The atomic COF structure from previous work was adopted for COF monolayer. Five layers were stacked together in an ABA manner with an interlayer spacing of 0.34 nm to form the cylindrical nanochannels. The atomic charges of COF atoms were calculated by DFT method with the PBE0 and def2tzvp basis set using Gaussian 16. The TIP4P/2005 model was used for water molecules, and the force field was used for other ions, including K$^+$, Na$^+$, Li$^+$, and Cl$^-$. The Shake algorithm was used to maintain the rigidity of water molecules. Lennard-Jones (LJ) potential was cut and shifted at 1.0 nm. The LJ potential parameters for COF atoms were taken from the Dreiding force field. All simulations were carried out using the parallel MD software package LAMMPS. Periodic boundary conditions were imposed in all directions. The velocity-verlet algorithm with a time step of 2 fs was used to integrate the equation of motion, and a Nosé–Hoover thermostat with a time constant of 200 fs was used to maintain the temperature of fluid at $T = 298$ K. A Nosé–Hoover barostat with a time constant of 2 ps was applied to maintain the fluid pressure along z-direction at p = 1 atm. During simulation, all atoms of COF layers were frozen at their initial positions.

## Data availability

All data supporting the findings of this study are available within the article and the Supplementary Information file, or available from the corresponding authors upon request. Source data are provided with this paper.

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

## Acknowledgements

The authors gratefully acknowledge financial support from National Key Research and Development Program of China (No. 2021YFB3802200, by F.P.), National Natural Science Foundation of China (No. 21838008, by Z.J.), Program of Introducing Talents of Discipline to Universities (No. BP0618007, by Z.J.), Open Project of The National Laboratory of Solid State Microstructures, Nanjing University (No. M33028, by K.Z.) and Key Laboratory of Special Functional Materials and Structural Design of Ministry of Education (Class B), Lanzhou University (No. lzujbky-2021-kb06, by K.Z.).

## Author contributions

Z.J. and F.P. conceived the idea and designed the research. H.W. carried out the experiment. H.W., Y.L., and B.S. synthesized COF nanosheets. K.Z. and Y.Z. carried out structural characterizations. Y.L. (Yawei Liu) and Y.C. carried out molecular dynamic simulations. R.L. provided constructive suggestions for data visualization. Z.Z., H.J., Z.G., M.W., and L.C. provided constructive suggestions for results and discussion. All authors participated in the discussion. Z.J., F.P., H.W., and K.Z. co-wrote the manuscript.

## Competing interests

The authors declare no competing interests.
