## [Peer Review File · Nature Communications]

Covalent Organic Framework Membranes for Efficient Separation of Monovalent CationsReviewers' Comments:

Reviewer #1:

Remarks to the Author:

It is a nice article written by experts in the field. I recommend publication after minor revision:

The authors say in ABSTRACT "The resulting COF membrane acquired an actual selectivity of 4.2-4.7 for K⁺/Li⁺ binary mixtures, an ideal selectivity of ~13.7 for K⁺/Li⁺, superior to ever reported membranes." Is this statement really true? In the paper "Ion separations with membranes" by Chao Tang and Merlin L. Bruening, these authors say: "With GO membranes that have a 9.8 Å interlayer spacing, monovalent cations diffuse through the membrane two orders of magnitude faster than Ca²⁺ and 3-4 orders of magnitudes faster than Mg²⁺. Further reducing the interlayer spacing to 8.6 Å in GO membranes leads to K⁺/Li⁺ and Na⁺/Li⁺ selectivities close to 100...". At another position, Tang and Bruening state "For example, Guo and coworkers fabricated a PSS-threaded MOF membrane and found Li⁺/Na⁺, Li⁺/K⁺, and Li⁺/Mg²⁺ selectivities of 35, 67, and 1,815...". That there is a big difference between the ideal and the actual selectivity is known: These membranes exhibited K⁺/Na⁺ selectivities as large as 84 in single-salt conductivity measurements, however, the selectivities dropped to 5 when K⁺ and Na⁺ are both present in the mixture."(E. T. Acar, S. F. Buchsbaum, C. Combs, F. Fornasiero, Z. S. Siwy, *Sci. Adv.* 2019, 5, eaav2568).

Authors, discuss more critically the performance of COFs in ion sieving. Discuss more clearly the benefits of COFs, maybe in the trade-off between selectivity and permeability/permeance.

Minor remarks:

- In Fig. 6D, the K⁺-permeance is given in [m²/s]. This is the unit of a diffusivity. Permeance is mol per area and time and pressure difference.
- In Fig. 4 is a mix of the units given in [...] and (...)
- In Fig. 6, you mix also [...] and (...).
- Sometimes there is space, sometimes not. Example: Fig. 6C without space T(K) but in 6E with space r [nm]

Reviewer #2:

Remarks to the Author:

Extracting and purifying monovalent cations attracts great attentions owing to the growing demand in energy storage materials. The separation of mono-/bi-valent cations has made great progress in recent year, in contrast, much less effort has been devoted to the separation of monovalent cation mixtures because of more severe challenges. In this study, for the first time, the authors demonstrated the utilization of covalent organic framework (COF) membranes in monovalent cation separation. They proposed a novel concept, confined cascade separation, to engineer COF membranes with high separation efficiency. The resulting COF membranes exhibited the ever-reported highest selectivity in separation of binary mixtures of monovalent cations. This work opens a new avenue to rational design and preparation of high-performance ionic separation membranes, and should be of great interest to researchers in difference scientific communities such as porous organic materials and chemical separations. To be frank, I much enjoyed reading this manuscript. The experiment, characterization and simulation results on membrane structures and mass transport mechanisms are well organized and extensively discussed. No doubt, the breakthrough in separation of monovalent cations in this study will stimulate the thinking about the disruptive membrane materials. Overall, this manuscript well meets the scope and high standard of Nature Communications, and should be accepted after addressing the following minor issues.

1. The fabricated COF membranes exhibited a smooth, defect-free and tight morphologies, which is different from those from other two-dimensional membranes, such Mxene, MoS₂ and GO membranes. Please discuss what makes such difference.
2. The experimental XRD pattern and 2D-GIWAXS pattern were highly consistent with the simulated

reversed AA stacking XRD patterns as indicated by the existence of the sharp diffraction peaks corresponding to the 100 planes. Which crystal planes do the other peaks correspond to? The stack orientation of crystal planes should be clarified.

3. This study demonstrated a kind of hydrophilic COF membranes with high water uptake. However, in principle, the higher water uptake usually causes higher swelling ratio, originating from the flexible polymer networks nature. However, the COF membrane in this study kept stable performances during aqueous operations. It is recommended to provide more detailed explanation on the water uptake and anti-swelling properties of the resulting membranes.

4. Why do TpPa-SO₃H membranes with thicker hydration shell had lower sieving performances than TpPa-PO₃H₂ membranes? Please clarify.

5. Please unify whether the jargon is "selective ion transport" or "selective cation transport", and check the writing errors in the whole manuscript.

6. Please check supplementary Table S6

7. Supplementary figure S22b presented the separation performance under different temperature. The activation energy is suggested.

Some minor issues

1. P. 9, line 157, Fig. S13...

2. P. 18, line 315, Fig. S18...

Reviewers' Comments:

Reviewer #1 (Remarks to the Author):

It is a nice article written by experts in the field. I recommend publication after minor revision:

The authors say in ABSTRACT “The resulting COF membrane acquired an actual selectivity of 4.2-4.7 for K⁺/Li⁺ binary mixtures, an ideal selectivity of ~13.7 for K⁺/Li⁺, superior to ever reported membranes.” Is this statement really true? In the paper “Ion separations with membranes” by Chao Tang and Merlin L. Bruening, these authors say: “With GO membranes that have a 9.8 Å interlayer spacing, monovalent cations diffuse through the membrane two orders of magnitude faster than Ca²⁺ and 3–4 orders of magnitudes faster than Mg²⁺. Further reducing the interlayer spacing to 8.6 Å in GO membranes leads to K⁺/Li⁺ and Na⁺/Li⁺ selectivities close to 100...”. At another position, Tang and Bruening state “For example, Guo and coworkers fabricated a PSS-threaded MOF membrane and found Li⁺/Na⁺, Li⁺/K⁺, and Li⁺/Mg²⁺ selectivities of 35, 67, and 1,815...”. That there is a big difference between the ideal and the actual selectivity is known: These membranes exhibited K⁺/Na⁺ selectivities as large as 84 in single-salt conductivity measurements, however, the selectivities dropped to 5 when K⁺ and Na⁺ are both present in the mixture.”(E. T. Acar, S. F. Buchsbaum, C. Combs, F. Fornasiero, Z. S. Siwy, *Sci. Adv.* 2019, 5, eaav2568).

Reply: Thank the reviewer for the highly positive remarks, kind encouragement and valuable guidance on our manuscript. We have made thorough revisions on our original manuscript accordingly. First of all, we are sorry for our too subjective claim. Base on the reviewer’s guidance, we conducted further surveys on the research work about ion separations with membrane technology. We have revised the relevant description as shown below, and supplemented the related references including the above references in the revised manuscript.:

Abstract: “The resulting COF membrane acquired good ion separation performances, having an actual selectivity of 4.2-4.7 for K⁺/Li⁺ binary mixtures, an ideal selectivity of ~13.7 for K⁺/Li⁺.”

Introduction: “The overwhelming majority of researches deal with single ion solution and the mono/monovalent cation selectivity is evaluated by the ideal selectivity, which is calculated from the ratio of individual permeation flux of single ion and usually lies within 2-100⁷⁻¹³. Few researches deal with binary mono/monovalent cation mixtures using graphene oxide and metal organic framework membranes¹⁴⁻¹⁸, and

the acquired selectivity is much lower than the ideal selectivity.”

Discussion: “By manipulating the channel properties of COF membranes, our *de novo* designed phosphoric COF (TpPa-PO₃H₂) membrane exhibited good separation performance for monovalent cations, with an actual selectivity of 4.2-4.7 for K⁺/Li⁺ binary mixtures and an ideal selectivity of ~13.7 for K⁺/Li⁺.”

Moreover, we supplemented five latest publications in the revised manuscript.

Added reference:

7 Tang, C. & Bruening, M. L. Ion separations with membranes. *Journal of Polymer Science* **58**, 2831-2856 (2020).

13 Abraham, J. *et al.* Tunable sieving of ions using graphene oxide membranes. *Nature Nanotechnology* **12**, 546-550 (2017).

14 Guo, Y., Ying, Y., Mao, Y., Peng, X. & Chen, B. Polystyrene Sulfonate Threaded through a Metal–Organic Framework Membrane for Fast and Selective Lithium-Ion Separation. *Angew. Chem. Int. Ed.* **55**, 15120-15124 (2016).

17 Zhang, H. *et al.* Ultrafast selective transport of alkali metal ions in metal organic frameworks with subnanometer pores. *Science Advances* **4**, eaaq0066.

18 Acar, E. T., Buchsbaum, S. F., Combs, C., Fornasiero, F. & Siwy, Z. S. Biomimetic potassium-selective nanopores. *Science Advances* **5**, eaav2568.

Authors, discuss more critically the performance of COFs in ion sieving. Discuss more clearly the benefits of COFs, maybe in the trade-off between selectivity and permeability/permeance.

Reply: According to the reviewer’s valuable guidance, we had added more discussions about the performance of COFs in ion sieving, as shown below:

Introduction: “The atomically smooth channels of COFs are beneficial for the rapid transport of penetrating ions. The long-range ordered structures impart COF membranes narrow distribution of channel size, ensuring excellent ion sieving effect^{22,23}. The channel size of COFs can be finely tailored in the range of 0.5-5.0 nm through the rational screening of linker and linkages²⁴, which provides a high freedom for membrane construction toward target ion separations. Moreover, COFs possess abundant monodispersed functional groups, thus more easily incorporating multiple physicochemical interactions for high recognition ability toward target ions²⁵. These features render COF membranes a wealth of chances for

surpassing the trade-off limitation between permeability and selectivity in ion separations. However, because the channel size in COF membranes is usually much larger than the ion sizes, only two reports^{26,27} concerning COF membranes for mono/bivalent ion separation can be found so far. It therefore anticipates great breakthrough to use COF membranes for achieving efficient ion separations.”

Minor remarks:

1. In Fig. 6D, the K⁺-permeance is given in [m²/s]. This is the unit of a diffusivity. Permeance is mol per area and time and pressure difference.

Reply: Thank the reviewer for the guidance, the unit of K⁺ in Fig. 6D has been revised to diffusivity.

2. In Fig. 4 is a mix of the units given in [...] and (...).

Reply: The units in Fig. 4 have been revised as (...).

3. In Fig. 6, you mix also [...] and (...).

Reply: The units in Fig. 6 have been revised as (...).

4. Sometimes there is space, sometimes not. Example: Fig. 6C without space T(K) but in 6E with space r [nm].

Reply: Thank the reviewer for the guidance. We have added space between character and unit in Fig. 6c. Moreover, we conducted a thorough check on the units and legends in the revised manuscript and supplementary information.

Reviewer: 2

Extracting and purifying monovalent cations attracts great attentions owing to the growing demand in energy storage materials. The separation of mono-/bi-valent cations has made great progress in recent year, in contrast, much less effort has been devoted to the separation of monovalent cation mixtures because of more severe challenges. In this study, for the first time, the authors demonstrated the utilization of covalent organic framework (COF) membranes in monovalent cation separation. They proposed a novel concept, confined cascade separation, to engineer COF membranes with high separation efficiency. The resulting COF membranes exhibited the ever-reported highest selectivity in separation of binary mixtures of monovalent cations. This work opens a new avenue to rational design and preparation of high-performance ionic separation membranes, and should be of great interest to researchers in difference scientific communities such as porous organic materials and chemical separations. To be frank, I much enjoyed reading this manuscript. The experiment, characterization and simulation results on membrane structures and mass transport mechanisms are well organized and extensively discussed. No doubt, the breakthrough in separation of monovalent cations in this study will stimulate the thinking about the disruptive membrane materials. Overall, this manuscript well meets the scope and high standard of Nature Communications, and should be accepted after addressing the following minor issues.

Reply: Thank the reviewer for the highly positive remarks on our manuscript. We have made thorough revisions on our original manuscript and supporting information. The detailed responses are listed below.

1. The fabricated COF membranes exhibited a smooth, defect-free and tight morphologies, which is different from those from other two-dimensional membranes, such Mxene, MoS₂ and GO membranes. Please discuss what makes such difference.

Reply: In this work, a series of COF membranes with different thickness could be fabricated using vacuum assisted assembly method. The membrane shown in Fig. 2e has a dense and tight morphology with a thickness of $2.0 \pm 0.2 \mu\text{m}$. This cross-view morphology is similar to that of thin (less than 500 nm) graphene oxide membranes (*Science* **342**, 95-98 (2013); *Science* **361**, 52-57 (2018)). We attribute this phenomenon to the high aspect ratio (exceeding 1000 as shown in Fig. 2c-d) of COF nanosheets, which render enough contact sites for adjacent building blocks. Moreover, the abundant functional groups on COF nanosheets

also intensify the interaction between COF nanosheets, thus contributing to tight and dense structures. In sharp contrast, due to the difficulty in incorporating functional groups in other 2D materials, such as Mxene, MoS₂ and GO, the resulting membranes often had weaker interactions among 2D building blocks. Therefore, Mxene, MoS₂ and GO membranes usually exhibit typical micro-laminar cross-section views.

Accordingly, we revised our description in Results part, “Interestingly, the dense and uniform morphologies of COF layers were similar to that of thin (less than 500 nm) graphene oxide membranes^{29,30}. Such phenomenon could be attributed to the ultrahigh aspect ratio of COF nanosheets, which rendered contact sites for adjacent nanosheets. Moreover, the abundant functional groups on COF nanosheets also fortified the interactions among COF nanosheets, thus contributing to tight and dense structures of COF membranes.”

2. The experimental XRD pattern and 2D-GIWAXS pattern were highly consistent with the simulated reversed AA stacking XRD patterns as indicated by the existence of the sharp diffraction peaks corresponding to the 100 planes. Which crystal planes do the other peaks correspond to? The stack orientation of crystal planes should be clarified.

Reply: According to the reviewer’s valuable guidance, we have supplemented more description in the revised manuscript, including the crystal plane information and the stack orientation of crystal plane, as shown below: “X-ray diffraction (XRD) pattern showed that COF membranes had three signals appearing at $\sim 5.0^\circ$, $\sim 8.0^\circ$ and $\sim 27.5^\circ$ (Fig. 3b), which corresponded to (100) plane, (010) plane and (001) plane, respectively, agreeing well with the simulated patterns of eclipsed projections (Supplementary Fig. 13 and Supplementary Table 3). The high signal intensity of (100) plane around 5.0° manifested that the periodic 1D channel size of COF membranes had a concentration at ~ 14 Å. Owing to high crystallinity, the experimental channel size obtained from N₂ adsorption-desorption experiments was measured to be 13.3-13.4 Å (Fig. 3c, and Supplementary Fig. 14), in good accordance with the projected size of ~ 14 Å, ensuring long-range ordered channels and even-distributed acid groups along the 1D channels (Fig. 3d). Two-dimensional synchrotron radiation grazing incidence wide-angle X-ray scattering (2D-GIWAXS) spectra in Fig. 3e exhibited a diffraction projection signal of (100) plane emerging at $q_{xy} = 0.35$ Å⁻¹ and concentrating near $q_z = 0$ Å⁻¹, revealing a certain in-plane distribution of 1D channel perpendicular to the support for COF membranes. Moreover, it can be seen that a weak diffraction spot attributed to (001) plane

appeared at around $q_z = 1.9 \text{ \AA}^{-1}$ and concentrated near $q_{xy} = 0 \text{ \AA}^{-1}$, unveiling the horizontal orientation of π - π stack interlayers and high crystallinity of COF membranes.”

3. This study demonstrated a kind of hydrophilic COF membranes with high water uptake. However, in principle, the higher water uptake usually causes higher swelling ratio, originating from the flexible polymer networks nature. However, the COF membrane in this study kept stable performances during aqueous operations. It is recommended to provide more detailed explanation on the water uptake and anti-swelling properties of the resulting membranes.

Reply: Thank the reviewer for valuable guidance. In this study, the fabricated COF membranes exhibited high water uptake of 8.7, 8.4, and 6.7 $\text{mg}_{\text{water}}/\text{mg}_{\text{cof}}$ for TpPa-SO₃H, TpPa-PO₃H₂ and TpPa-CO₂H membranes, respectively (measured by QCM, shown in Fig 3G). The high water uptake could be attributed to two main factors: on one hand, COF membrane possessed high hydrophilicity, as indicated by the water contact angle of $\sim 29^\circ$, $\sim 39^\circ$ and $\sim 66^\circ$ for TpPa-SO₃H, TpPa-PO₃H₂ and TpPa-CO₂H membranes, respectively (Fig. 3F). The hydrophilicity imparted COF higher affinity toward water molecules. On the other hand, the large porosity rendered COFs large capacity to adsorb plenty of water molecules and exhibit high water uptake. To investigate aqueous stability performances, COF membranes were immersed into water for one month. TpPa-SO₃H, TpPa-PO₃H₂ and TpPa-CO₂H membranes only had low swelling degree below 5%. We attributed the superior aqueous stability to evenly distributed functional groups of COF nanosheets, which conferred abundant interaction sites (H-bonding and van der Waals force) among adjacent COF nanosheets, thus fortifying the structural stability of COF membranes in aqueous solution.

Accordingly, we revised the description in Results and Supplementary Materials parts, as shown below: Results: “The high hydrophilicity could fortify the interaction between COFs and water molecules. Moreover, the porous crystalline nature afforded more interaction sites for COFs and water molecules, thus contributing to large water adsorption uptake as measured by quartz crystal microbalance (QCM). The water capacity of TpPa-PO₃H₂ and TpPa-SO₃H membranes was over 20% than that of TpPa-CO₂H, reaching 8.4 mg/mg and 8.7 mg/mg , respectively (Fig. 3g). The charged property further supported the water affinity of COF membranes. Zeta potential results in Supplementary Table 4 showed a value of $-131.2 \pm 1.2 \text{ mV}$, $-129.3 \pm 3.1 \text{ mV}$ and $-64.0 \pm 0.3 \text{ mV}$ for TpPa-SO₃H, TpPa-PO₃H₂ and TpPa-CO₂H membranes, respectively. The facile dissociation of -SO₃H and -PO₃H₂ groups endowed the membranes

with abundant sites to tightly bind water molecules. In contrast, $-\text{CO}_2\text{H}$ group with weak dissociation showed lower charge property and hydrophilicity. Apart from moderate water affinity and uptake, COF membranes had excellent aqueous stability, only showing small swelling degree less than 5% during long-term test (Supplementary Table 5).”

Supplementary: “After immersing into aqueous conditions for one month, the COF membranes did not show macroscopic change or obvious macroscopic degradations. We assessed the swelling degree of COF membranes *via* the volume change before and after aqueous condition treatment, as shown in Table S5. The COF membranes only showed swelling degree $< 5\%$, much smaller than that of pristine GO membranes (60%-70%) and pristine MoS_2 membranes (5%-10%). This was mainly due to evenly distributed functional groups of COF nanosheets, which conferred abundant interaction sites (H-bonding and van der Waals force) among adjacent COF nanosheets, thus fortifying the structural stability of COF membranes, conferring COF membranes grand potentials for long-term utilization in aqueous solution.”

4. Why do TpPa-SO₃H membranes with thicker hydration shell had lower sieving performances than TpPa-PO₃H₂ membranes? Please clarify.

Reply: In this study, we proposed confined cascade separation concept to elucidate the ion transport behavior within COF membranes. The acid-domains with $-\text{PO}_3\text{H}_2$ and $-\text{SO}_3\text{H}$ groups, serving as “confined stage”, can sufficiently bind water molecules to form hydration shells, narrowing the effective channel size between the size of target cations. Subsequently, a number of the acid-domains along channels of COF membrane, amplified the sieving function of each “stage”, rendering high selectivity for monovalent cations. A set of descriptors, i.e., hydration energy, group density, group distance, were designed to determine the “stage” properties. The results in Fig. 5c-5f illustrated that hydration energy, acid group density and acid group distance have synergistic effects on ionic transport. TpPa-SO₃H and TpPa-PO₃H₂ membranes had higher ion selectivity than TpPa-CO₂H membrane owing to moderate hydration shell thickness and intrinsic channel size. For monovalent ion sieving, TpPa-SO₃H and TpPa-PO₃H₂ membranes also exhibited actual selectivity of 2.56 and 4.21, respectively, which were higher than that of TpPa-CO₂H membrane. Particularly, it should note that for the separation of binary system, the monovalent cations (e.g., K^+ and Li^+) tend to interact with acid groups in COFs and had a competition with water. As a result, the stability and continuity of hydration shell formed within COFs would be leveraged by monovalent cations. Although

a thicker hydration shell in pure water could be formed in TpPa-SO₃H membrane (Fig. 4), the corresponding stability of hydration shell would be challenged by penetrating cations, thus weakening the ion sieving performance of membrane. In contrast, -PO₃H₂ could well retain the original hydration shell during separation, which was more favorable for accomplishing the sieving property of hydration shell than -SO₃H, thus coffering a higher ion sieving performance to TpPa-PO₃H₂ membrane.

5. Please unify whether the jargon is “selective ion transport” or “selective cation transport”, and check the writing errors in the whole manuscript.

Reply: The jargon has been unified as “selective ion transport” and the writing errors in the manuscript has been revised.

6. Please check supplementary Table S6.

Reply: The format of Supplementary Table 6 has been revised.

7. Supplementary figure S22b presented the separation performance under different temperature. The activation energy is suggested.

Reply: The activation energy for Fig. 6c and Supplementary Fig. 22b has been added, as shown in Supplementary Fig. 22c for TpPa-SO₃H membrane and Supplementary Fig. 22d for TpPa-PO₃H₂ membrane, respectively. The corresponding descriptions have been added in Results and Supplementary information.

Results: “Compared with TpPa-PO₃H₂ membrane, TpPa-SO₃H membrane showed a more obvious change on permeation rates with the varied temperature. This was mainly due to the activity of -SO₃H group was more sensitive than -PO₃H₂ group under different temperature (Supplementary Fig. 22c-d).”

Supplementary information: “The activation energy of K⁺ and Li⁺ was 10.94 kJ mol⁻¹ and 13.01 kJ mol⁻¹ for TpPa-SO₃H membrane, which was higher than K⁺ and Li⁺ for TpPa-PO₃H₂ membrane (5.80 kJ mol⁻¹ and 7.83 kJ mol⁻¹), as shown in Supplementary Fig. 24c-d. This result demonstrated that -SO₃H group was more sensitive than -PO₃H₂ group toward different temperature, which led to TpPa-SO₃H membrane had a more significant decrease of permeation rate with the increased operation temperature.”

8. Some minor issues: 1. P. 9, line 157, Fig. S13; 2. P. 18, line 315, Fig. S18.

Reply: “Fig 13” in P.9, line 157 has been revised to “Fig. 13”. “Fig S18” in P.18, line 315 has been revised to “Supplementary Fig. 18”.

Reviewers' Comments:

Reviewer #1:

Remarks to the Author:

I recommend acceptance.

Nevertheless, there are some minor editorial questions:

Fig. 1: Neither in figure heading nor in the text, the j is explained. Also " $2 \rightarrow j-2$ " and " $j+1 \rightarrow j-1$ ".

Fig. 5: "Selective transport of monovalent cations within COF membranes." Better: ... through COF membranes

Fig. 6: "Separation performances for binary mixtures of monovalent cations within COF membranes".

Better: ... through COF membranes

Line 76: ... 1) the channels... but Line 80: ... 2) All acid-domains... Go through the whole text and unify how to continue after double point : with uppercase (recommended) or lowercase.

Reviewer #2:

Remarks to the Author:

The authors have made proper modifications to their manuscript, and I am satisfied with the revised version. It can be accepted for publication now.

Reviewers' Comments:

The first-round reviewing

Reviewer #1 (Remarks to the Author):

It is a nice article written by experts in the field. I recommend publication after minor revision:

The authors say in ABSTRACT “The resulting COF membrane acquired an actual selectivity of 4.2-4.7 for K⁺/Li⁺ binary mixtures, an ideal selectivity of ~13.7 for K⁺/Li⁺, superior to ever reported membranes.” Is this statement really true? In the paper “Ion separations with membranes” by Chao Tang and Merlin L. Bruening, these authors say: “With GO membranes that have a 9.8 Å interlayer spacing, monovalent cations diffuse through the membrane two orders of magnitude faster than Ca²⁺ and 3–4 orders of magnitudes faster than Mg²⁺. Further reducing the interlayer spacing to 8.6 Å in GO membranes leads to K⁺/Li⁺ and Na⁺/Li⁺ selectivities close to 100...”. At another position, Tang and Bruening state “For example, Guo and coworkers fabricated a PSS-threaded MOF membrane and found Li⁺/Na⁺, Li⁺/K⁺, and Li⁺/Mg²⁺ selectivities of 35, 67, and 1,815...”. That there is a big difference between the ideal and the actual selectivity is known: These membranes exhibited K⁺/Na⁺ selectivities as large as 84 in single-salt conductivity measurements, however, the selectivities dropped to 5 when K⁺ and Na⁺ are both present in the mixture.”(E. T. Acar, S. F. Buchsbaum, C. Combs, F. Fornasiero, Z. S. Siwy, *Sci. Adv.* 2019, 5, eaav2568).

Reply: Thank the reviewer for the highly positive remarks, kind encouragement and valuable guidance on our manuscript. We have made thorough revisions on our original manuscript accordingly. First of all, we are sorry for our too subjective claim. Base on the reviewer’s guidance, we conducted further surveys on the research work about ion separations with membrane technology. We have revised the relevant description as shown below, and supplemented the related references including the above references in the revised manuscript.:

Abstract: “The resulting COF membrane acquired good ion separation performances, having an actual selectivity of 4.2-4.7 for K⁺/Li⁺ binary mixtures, an ideal selectivity of ~13.7 for K⁺/Li⁺.”

Introduction: “The overwhelming majority of researches deal with single ion solution and the mono/monovalent cation selectivity is evaluated by the ideal selectivity, which is calculated from the ratio of individual permeation flux of single ion and usually lies within 2-100⁷⁻¹³. Few researches deal with binary mono/monovalent cation mixtures using graphene oxide and metal organic framework membranes¹⁴⁻¹⁸, and

the acquired selectivity is much lower than the ideal selectivity.”

Discussion: “By manipulating the channel properties of COF membranes, our *de novo* designed phosphoric COF (TpPa-PO₃H₂) membrane exhibited good separation performance for monovalent cations, with an actual selectivity of 4.2-4.7 for K⁺/Li⁺ binary mixtures and an ideal selectivity of ~13.7 for K⁺/Li⁺.”

Moreover, we supplemented five latest publications in the revised manuscript.

Added reference:

7 Tang, C. & Bruening, M. L. Ion separations with membranes. *Journal of Polymer Science* **58**, 2831-2856 (2020).

13 Abraham, J. *et al.* Tunable sieving of ions using graphene oxide membranes. *Nature Nanotechnology* **12**, 546-550 (2017).

14 Guo, Y., Ying, Y., Mao, Y., Peng, X. & Chen, B. Polystyrene Sulfonate Threaded through a Metal–Organic Framework Membrane for Fast and Selective Lithium-Ion Separation. *Angew. Chem. Int. Ed.* **55**, 15120-15124 (2016).

17 Zhang, H. *et al.* Ultrafast selective transport of alkali metal ions in metal organic frameworks with subnanometer pores. *Science Advances* **4**, eaaq0066.

18 Acar, E. T., Buchsbaum, S. F., Combs, C., Fornasiero, F. & Siwy, Z. S. Biomimetic potassium-selective nanopores. *Science Advances* **5**, eaav2568.

Authors, discuss more critically the performance of COFs in ion sieving. Discuss more clearly the benefits of COFs, maybe in the trade-off between selectivity and permeability/permeance.

Reply: According to the reviewer’s valuable guidance, we had added more discussions about the performance of COFs in ion sieving, as shown below:

Introduction: “The atomically smooth channels of COFs are beneficial for the rapid transport of penetrating ions. The long-range ordered structures impart COF membranes narrow distribution of channel size, ensuring excellent ion sieving effect^{22,23}. The channel size of COFs can be finely tailored in the range of 0.5-5.0 nm through the rational screening of linker and linkages²⁴, which provides a high freedom for membrane construction toward target ion separations. Moreover, COFs possess abundant monodispersed functional groups, thus more easily incorporating multiple physicochemical interactions for high recognition ability toward target ions²⁵. These features render COF membranes a wealth of chances for

surpassing the trade-off limitation between permeability and selectivity in ion separations. However, because the channel size in COF membranes is usually much larger than the ion sizes, only two reports^{26,27} concerning COF membranes for mono/bivalent ion separation can be found so far. It therefore anticipates great breakthrough to use COF membranes for achieving efficient ion separations.”

Minor remarks:

1. In Fig. 6D, the K⁺-permeance is given in [m²/s]. This is the unit of a diffusivity. Permeance is mol per area and time and pressure difference.

Reply: Thank the reviewer for the guidance, the unit of K⁺ in Fig. 6D has been revised to diffusivity.

2. In Fig. 4 is a mix of the units given in [...] and (...).

Reply: The units in Fig. 4 have been revised as (...).

3. In Fig. 6, you mix also [...] and (...).

Reply: The units in Fig. 6 have been revised as (...).

4. Sometimes there is space, sometimes not. Example: Fig. 6C without space T(K) but in 6E with space r [nm].

Reply: Thank the reviewer for the guidance. We have added space between character and unit in Fig. 6c. Moreover, we conducted a thorough check on the units and legends in the revised manuscript and supplementary information.

Reviewer: 2

Extracting and purifying monovalent cations attracts great attentions owing to the growing demand in energy storage materials. The separation of mono-/bi-valent cations has made great progress in recent year, in contrast, much less effort has been devoted to the separation of monovalent cation mixtures because of more severe challenges. In this study, for the first time, the authors demonstrated the utilization of covalent organic framework (COF) membranes in monovalent cation separation. They proposed a novel concept, confined cascade separation, to engineer COF membranes with high separation efficiency. The resulting COF membranes exhibited the ever-reported highest selectivity in separation of binary mixtures of monovalent cations. This work opens a new avenue to rational design and preparation of high-performance ionic separation membranes, and should be of great interest to researchers in difference scientific communities such as porous organic materials and chemical separations. To be frank, I much enjoyed reading this manuscript. The experiment, characterization and simulation results on membrane structures and mass transport mechanisms are well organized and extensively discussed. No doubt, the breakthrough in separation of monovalent cations in this study will stimulate the thinking about the disruptive membrane materials. Overall, this manuscript well meets the scope and high standard of Nature Communications, and should be accepted after addressing the following minor issues.

Reply: Thank the reviewer for the highly positive remarks on our manuscript. We have made thorough revisions on our original manuscript and supporting information. The detailed responses are listed below.

1. The fabricated COF membranes exhibited a smooth, defect-free and tight morphologies, which is different from those from other two-dimensional membranes, such Mxene, MoS₂ and GO membranes. Please discuss what makes such difference.

Reply: In this work, a series of COF membranes with different thickness could be fabricated using vacuum assisted assembly method. The membrane shown in Fig. 2e has a dense and tight morphology with a thickness of $2.0 \pm 0.2 \mu\text{m}$. This cross-view morphology is similar to that of thin (less than 500 nm) graphene oxide membranes (*Science* **342**, 95-98 (2013); *Science* **361**, 52-57 (2018)). We attribute this phenomenon to the high aspect ratio (exceeding 1000 as shown in Fig. 2c-d) of COF nanosheets, which render enough contact sites for adjacent building blocks. Moreover, the abundant functional groups on COF nanosheets

also intensify the interaction between COF nanosheets, thus contributing to tight and dense structures. In sharp contrast, due to the difficulty in incorporating functional groups in other 2D materials, such as Mxene, MoS₂ and GO, the resulting membranes often had weaker interactions among 2D building blocks. Therefore, Mxene, MoS₂ and GO membranes usually exhibit typical micro-laminar cross-section views.

Accordingly, we revised our description in Results part, “Interestingly, the dense and uniform morphologies of COF layers were similar to that of thin (less than 500 nm) graphene oxide membranes^{29,30}. Such phenomenon could be attributed to the ultrahigh aspect ratio of COF nanosheets, which rendered contact sites for adjacent nanosheets. Moreover, the abundant functional groups on COF nanosheets also fortified the interactions among COF nanosheets, thus contributing to tight and dense structures of COF membranes.”

2. The experimental XRD pattern and 2D-GIWAXS pattern were highly consistent with the simulated reversed AA stacking XRD patterns as indicated by the existence of the sharp diffraction peaks corresponding to the 100 planes. Which crystal planes do the other peaks correspond to? The stack orientation of crystal planes should be clarified.

Reply: According to the reviewer’s valuable guidance, we have supplemented more description in the revised manuscript, including the crystal plane information and the stack orientation of crystal plane, as shown below: “X-ray diffraction (XRD) pattern showed that COF membranes had three signals appearing at $\sim 5.0^\circ$, $\sim 8.0^\circ$ and $\sim 27.5^\circ$ (Fig. 3b), which corresponded to (100) plane, (010) plane and (001) plane, respectively, agreeing well with the simulated patterns of eclipsed projections (Supplementary Fig. 13 and Supplementary Table 3). The high signal intensity of (100) plane around 5.0° manifested that the periodic 1D channel size of COF membranes had a concentration at $\sim 14 \text{ \AA}$. Owing to high crystallinity, the experimental channel size obtained from N₂ adsorption-desorption experiments was measured to be 13.3-13.4 \AA (Fig. 3c, and Supplementary Fig. 14), in good accordance with the projected size of $\sim 14 \text{ \AA}$, ensuring long-range ordered channels and even-distributed acid groups along the 1D channels (Fig. 3d). Two-dimensional synchrotron radiation grazing incidence wide-angle X-ray scattering (2D-GIWAXS) spectra in Fig. 3e exhibited a diffraction projection signal of (100) plane emerging at $q_{xy} = 0.35 \text{ \AA}^{-1}$ and concentrating near $q_z = 0 \text{ \AA}^{-1}$, revealing a certain in-plane distribution of 1D channel perpendicular to the support for COF membranes. Moreover, it can be seen that a weak diffraction spot attributed to (001) plane

appeared at around $q_z = 1.9 \text{ \AA}^{-1}$ and concentrated near $q_{xy} = 0 \text{ \AA}^{-1}$, unveiling the horizontal orientation of π - π stack interlayers and high crystallinity of COF membranes.”

3. This study demonstrated a kind of hydrophilic COF membranes with high water uptake. However, in principle, the higher water uptake usually causes higher swelling ratio, originating from the flexible polymer networks nature. However, the COF membrane in this study kept stable performances during aqueous operations. It is recommended to provide more detailed explanation on the water uptake and anti-swelling properties of the resulting membranes.

Reply: Thank the reviewer for valuable guidance. In this study, the fabricated COF membranes exhibited high water uptake of 8.7, 8.4, and 6.7 $\text{mg}_{\text{water}}/\text{mg}_{\text{cof}}$ for TpPa-SO₃H, TpPa-PO₃H₂ and TpPa-CO₂H membranes, respectively (measured by QCM, shown in Fig 3G). The high water uptake could be attributed to two main factors: on one hand, COF membrane possessed high hydrophilicity, as indicated by the water contact angle of $\sim 29^\circ$, $\sim 39^\circ$ and $\sim 66^\circ$ for TpPa-SO₃H, TpPa-PO₃H₂ and TpPa-CO₂H membranes, respectively (Fig. 3F). The hydrophilicity imparted COF higher affinity toward water molecules. On the other hand, the large porosity rendered COFs large capacity to adsorb plenty of water molecules and exhibit high water uptake. To investigate aqueous stability performances, COF membranes were immersed into water for one month. TpPa-SO₃H, TpPa-PO₃H₂ and TpPa-CO₂H membranes only had low swelling degree below 5%. We attributed the superior aqueous stability to evenly distributed functional groups of COF nanosheets, which conferred abundant interaction sites (H-bonding and van der Waals force) among adjacent COF nanosheets, thus fortifying the structural stability of COF membranes in aqueous solution.

Accordingly, we revised the description in Results and Supplementary Materials parts, as shown below: Results: “The high hydrophilicity could fortify the interaction between COFs and water molecules. Moreover, the porous crystalline nature afforded more interaction sites for COFs and water molecules, thus contributing to large water adsorption uptake as measured by quartz crystal microbalance (QCM). The water capacity of TpPa-PO₃H₂ and TpPa-SO₃H membranes was over 20% than that of TpPa-CO₂H, reaching 8.4 mg/mg and 8.7 mg/mg , respectively (Fig. 3g). The charged property further supported the water affinity of COF membranes. Zeta potential results in Supplementary Table 4 showed a value of $-131.2 \pm 1.2 \text{ mV}$, $-129.3 \pm 3.1 \text{ mV}$ and $-64.0 \pm 0.3 \text{ mV}$ for TpPa-SO₃H, TpPa-PO₃H₂ and TpPa-CO₂H membranes, respectively. The facile dissociation of -SO₃H and -PO₃H₂ groups endowed the membranes

with abundant sites to tightly bind water molecules. In contrast, $-\text{CO}_2\text{H}$ group with weak dissociation showed lower charge property and hydrophilicity. Apart from moderate water affinity and uptake, COF membranes had excellent aqueous stability, only showing small swelling degree less than 5% during long-term test (Supplementary Table 5).”

Supplementary: “After immersing into aqueous conditions for one month, the COF membranes did not show macroscopic change or obvious macroscopic degradations. We assessed the swelling degree of COF membranes *via* the volume change before and after aqueous condition treatment, as shown in Table S5. The COF membranes only showed swelling degree $< 5\%$, much smaller than that of pristine GO membranes (60%-70%) and pristine MoS_2 membranes (5%-10%). This was mainly due to evenly distributed functional groups of COF nanosheets, which conferred abundant interaction sites (H-bonding and van der Waals force) among adjacent COF nanosheets, thus fortifying the structural stability of COF membranes, conferring COF membranes grand potentials for long-term utilization in aqueous solution.”

4. Why do TpPa-SO₃H membranes with thicker hydration shell had lower sieving performances than TpPa-PO₃H₂ membranes? Please clarify.

Reply: In this study, we proposed confined cascade separation concept to elucidate the ion transport behavior within COF membranes. The acid-domains with $-\text{PO}_3\text{H}_2$ and $-\text{SO}_3\text{H}$ groups, serving as “confined stage”, can sufficiently bind water molecules to form hydration shells, narrowing the effective channel size between the size of target cations. Subsequently, a number of the acid-domains along channels of COF membrane, amplified the sieving function of each “stage”, rendering high selectivity for monovalent cations. A set of descriptors, i.e., hydration energy, group density, group distance, were designed to determine the “stage” properties. The results in Fig. 5c-5f illustrated that hydration energy, acid group density and acid group distance have synergistic effects on ionic transport. TpPa-SO₃H and TpPa-PO₃H₂ membranes had higher ion selectivity than TpPa-CO₂H membrane owing to moderate hydration shell thickness and intrinsic channel size. For monovalent ion sieving, TpPa-SO₃H and TpPa-PO₃H₂ membranes also exhibited actual selectivity of 2.56 and 4.21, respectively, which were higher than that of TpPa-CO₂H membrane. Particularly, it should note that for the separation of binary system, the monovalent cations (e.g., K^+ and Li^+) tend to interact with acid groups in COFs and had a competition with water. As a result, the stability and continuity of hydration shell formed within COFs would be leveraged by monovalent cations. Although

a thicker hydration shell in pure water could be formed in TpPa-SO₃H membrane (Fig. 4), the corresponding stability of hydration shell would be challenged by penetrating cations, thus weakening the ion sieving performance of membrane. In contrast, -PO₃H₂ could well retain the original hydration shell during separation, which was more favorable for accomplishing the sieving property of hydration shell than -SO₃H, thus coffering a higher ion sieving performance to TpPa-PO₃H₂ membrane.

5. Please unify whether the jargon is “selective ion transport” or “selective cation transport”, and check the writing errors in the whole manuscript.

Reply: The jargon has been unified as “selective ion transport” and the writing errors in the manuscript has been revised.

6. Please check supplementary Table S6.

Reply: The format of Supplementary Table 6 has been revised.

7. Supplementary figure S22b presented the separation performance under different temperature. The activation energy is suggested.

Reply: The activation energy for Fig. 6c and Supplementary Fig. 22b has been added, as shown in Supplementary Fig. 22c for TpPa-SO₃H membrane and Supplementary Fig. 22d for TpPa-PO₃H₂ membrane, respectively. The corresponding descriptions have been added in Results and Supplementary information.

Results: “Compared with TpPa-PO₃H₂ membrane, TpPa-SO₃H membrane showed a more obvious change on permeation rates with the varied temperature. This was mainly due to the activity of -SO₃H group was more sensitive than -PO₃H₂ group under different temperature (Supplementary Fig. 22c-d).”

Supplementary information: “The activation energy of K⁺ and Li⁺ was 10.94 kJ mol⁻¹ and 13.01 kJ mol⁻¹ for TpPa-SO₃H membrane, which was higher than K⁺ and Li⁺ for TpPa-PO₃H₂ membrane (5.80 kJ mol⁻¹ and 7.83 kJ mol⁻¹), as shown in Supplementary Fig. 24c-d. This result demonstrated that -SO₃H group was more sensitive than -PO₃H₂ group toward different temperature, which led to TpPa-SO₃H membrane had a more significant decrease of permeation rate with the increased operation temperature.”

8. Some minor issues: 1. P. 9, line 157, Fig. S13; 2. P. 18, line 315, Fig. S18.

Reply: “Fig 13” in P.9, line 157 has been revised to “Fig. 13”. “Fig S18” in P.18, line 315 has been revised to “Supplementary Fig. 18”.

Reviewers' Comments:

The second-round reviewing

Reviewer #1 (Remarks to the Author):

I recommend acceptance. Nevertheless, there are some minor editorial questions:

Reply: Thank the reviewer for the highly positive remarks on our manuscript. We have made revisions on our manuscript. The detailed responses are listed below.

1. Fig. 1: Neither in figure heading nor in the text, the j is explained. Also “2 to $j-2$ ” and “ $j+1$ to $j-1$ ”.

Reply: The corresponding explanation has been added in the figure heading, as shown in Fig. 1, Line 577: “($j-2$, $j-1$, j and $j+1$ mean the increasing numbers ranging between 1 and n)”.

2. Fig. 5: “Selective transport of monovalent cations within COF membranes.” Better: ... through COF membranes.

Reply: According to the reviewer's suggestion, the description “within COF membranes” in Fig. 5, Line 631 has been revised to “through COF membranes”.

3. Fig. 6: “Separation performances for binary mixtures of monovalent cations within COF membranes”. Better: ... through COF membranes.

Reply: The description “within COF membranes” in Fig. 6, Line 653 has been revised to “through COF membranes”.

4. Line 76: ... 1) the channels... but Line 80: ... 2) All acid-domains... Go through the whole text and unify how to continue after double point : with uppercase (recommended) or lowercase.

Reply: The text after double point has been revised with uppercase format, as shown in Line 75 “The

channels” and Line 81 “The acid-free-domains”.

Reviewer #2 (Remarks to the Author):

The authors have made proper modifications to their manuscript, and I am satisfied with the revised version.

It can be accepted for publication now.

Reply: Thank the reviewer for the highly positive comments and the efforts in reviewing this manuscript.